# Convergence of Shallow ReLU Networks on Weakly Interacting Data

**Léo Dana** [*]
Sierra, INRIA
Paris

**Loucas Pillaud-Vivien** [†]
CERMICS, Ecole Nationale des Ponts et Chaussées
Champs-sur-Marne

**Francis Bach** [‡]
Sierra, INRIA
Paris

## Abstract

We analyse the convergence of one-hidden-layer ReLU networks trained by gradient flow on $n$ data points. Our main contribution leverages the high dimensionality of the ambient space, which implies low correlation of the input samples, to demonstrate that a network with width of order $\log(n)$ neurons suffices for global convergence with high probability. Our analysis uses a Polyak–Łojasiewicz viewpoint along the gradient-flow trajectory, which provides an exponential rate of convergence of $\frac{1}{n}$. When the data are exactly orthogonal, we give further refined characterizations of the convergence speed, proving its asymptotic behavior lies between the orders $\frac{1}{n}$ and $\frac{1}{\sqrt{n}}$, and exhibiting a phase-transition phenomenon in the convergence rate, during which it evolves from the lower bound to the upper, and in a relative time of order $\frac{1}{\log(n)}$.

## 1 Introduction

Understanding the properties of models used in machine learning is crucial for providing guarantees to downstream users. Of particular importance, the convergence of the training process under gradient methods stands as one of the first issues to address in order to comprehend them. If, on the one hand, such a question for linear models and convex optimization problems [Bottou et al., 2018, Bach, 2024] are well understood, this is not the case for neural networks, which are the most used models in large-scale machine learning. This paper focuses on providing quantitative convergence guarantees for a one-hidden-layer neural network.

Theoretically, such global convergence analysis of neural networks has seen two main achievements in the past years: (i) the identification of the *lazy regime*, due to a particular initialization, where convergence is always guaranteed at the cost of being essentially a linear model [Jacot et al., 2018, Arora et al., 2019, Chizat et al., 2019], and (ii) the proof that with an infinite amount of hidden units a two-layer neural network converges towards the global minimizer of the loss [Mei et al., 2018, Chizat and Bach, 2018, Rotskoff and Vanden-Eijnden, 2018]. However, neural networks are trained in practice outside of these regimes, as neural networks are known to perform *feature learning*, and experimentally reach global minimum with a large but finite number of neurons. Quantifying in which regimes neural networks converge to a global minimum of their loss is still an important open question.

We identify a regime—marked by low-correlated inputs—where the training dynamics of shallow neural networks via gradient flow can be rigorously understood. Unlike prior analyses that hinged on finely tuned initialization scales [Chizat et al., 2019, Boursier et al., 2022], an infinite number of neurons [Jacot et al., 2018, Chizat and Bach, 2018], or the unrealistic orthogonality of data [Boursier

---

[*] `leo.dana@inria.fr`

[†] `loucas.pillaud-vivien@enpc.fr`

[‡] `francis.bach@inria.fr`

39th Conference on Neural Information Processing Systems (NeurIPS 2025).

et al., 2022, Frei et al., 2023], our setting arises naturally in high dimensions, notably when the input dimension $d$ exceeds $n^2$. Beyond existence of convergence, we seek to quantify it: how fast does the system lock onto a global minimizer? What governs this speed? Our work provides sharp answers.

We summarize our contributions in the analysis of the learning dynamics of a one-hidden-layer ReLU network on a finite number of data $n$ via gradient flow.

- Our main contribution is that the gradient flow training of shallow neural networks, with square error, on $n$ low correlated input, converges globally, i.e. converges to a neural network that interpolates exactly the data. We show that this occurs with high probability for high dimensional whitened input data as soon as $d \gtrsim n^2$. Furthermore, this convergence occurs for any initialization scale and whenever the neural network has more that $\log(n)$ neurons. We also show that the loss converges to zero exponentially fast with a rate at least of order $\frac{1}{n}$.

- Then, when the inputs are orthogonal, we refine our analysis in order to characterize the range of possible asymptotic speeds, which we find to be at most of order $\frac{1}{\sqrt{n}}$. Moreover, we conjecture that this speed is always of the highest order $\frac{1}{\sqrt{n}}$ with high probability and verify empirically this claim.

- Finally, for orthonormal inputs and a special initialization of the network, we highlight a phase transition in the convergence rate during the system's evolution, and compute the associated cut-off time and transition period.

## 2  Problem Setup

**Notations.**  We use $||v||$ to denote the euclidean norm of a vector $v$, $\langle \cdot | \cdot \rangle$ its scalar product, and $||M||$ for the operator norm associated with $|| \cdot ||$ of a matrix $M$. Moreover, let $\bar{v} = \frac{v}{||v||}$.

**Loss function.**  Let $(x_i, y_i)_{i=1:n} \in (\mathbb{R}^d \times \mathbb{R})^n$ be a sample of input vectors and real outputs. Let $d \in \mathbb{N}^*$ be the dimension of the vector space and $n \in \mathbb{N}^*$ the number of data points. In order to learn the regression problem of mapping $x_i$ to $y_i$, we use one-hidden-layer ReLU neural networks, which we write:

$$h_\theta(x) = \frac{1}{p} \sum_{j=1}^p a_j \sigma(\langle w_j | x \rangle) \,, \tag{1}$$

where $p \in \mathbb{N}^*$ is the number of units, $\sigma(x) = \max\{0, x\}$ for $x \in \mathbb{R}$ is the rectified linear unit (ReLU), and the parameters are gathered in $\theta = (a_j, w_j)_{1 \leq j \leq p} \in (\mathbb{R} \times \mathbb{R}^d)^p$. To simplify the ReLU notation, we define $\sigma(\langle w_j | x_i \rangle) = \langle w_j | x_i \rangle_+$ and $\mathbb{1}_{\langle w_j | x_i \rangle > 0} = 1_{j,i}$. When mentioning neurons of the network, we refer to $\langle w_j | x_i \rangle_+$, while second layer neurons refer to $a_j$. Neurons can be activated if $\langle w_j | x_i \rangle_+ > 0$, and are correctly activated if moreover $a_j y_i > 0$. Upon this prediction class and data, we analyse the regression loss with square error,

$$L(\theta) := \frac{1}{2n} \sum_{i=1}^n (y_i - h_\theta(x_i))^2 \,. \tag{2}$$

As soon as $d \geq n$, $(x_i)_{i=1:n}$ can form a free family, in which case the set of minima of $L$, which consists of all interpolators, is non-empty. We note $r_i = y_i - h_\theta(x_i)$ the residual of the loss.

**Gradient flow.**  In order to understand a simplified version of the optimization dynamics of this neural network, we study the continuous-time limit of gradient descent. We initialize $\theta_{t=0} = \theta_0$ and follow for all $t \geq 0$ the ordinary differential equation

$$\frac{d}{dt} \theta_t = -p \nabla_{\theta_t} L(\theta_t) \,, \tag{3}$$

where we choose a particular element of the sub-differential of the ReLU $\sigma'(x) = \mathbb{1}_{x>0}$, for any $x \in \mathbb{R}$. This choice is motivated by both prior empirical work from Bertoin et al. [2021] and theoretical work from Boursier et al. [2022, Proposition 2] and Jentzen and Riekert [2023]. Because ReLU is not differentiable at 0, we don't have a unique valid trajectory satisfying the gradient flow

equation. We thus chose among all the trajectories the only one for which deactivated neurons cannot reactivate themselves alone (more in Appendix C.2). We also decided to accelerate the dynamics by a factor $p$ as only this scaling gives a consistent mean field limit for the gradient flow when the number of neurons tends to infinity (see Definition 2.2 by Chizat and Bach [2018]).

**Weight invariance.** The 1-homogeneity of the ReLU provides a continuous symmetry in the function $\theta \mapsto h_\theta$ and hence the loss[4]. This feature is known to lead automatically to invariants in the gradient flow as explained generally by Marcotte et al. [2024]. The following lemma is not new [Wojtowytsch, 2020, p.11], and shows that, from this invariance, we deduce that the two layers have balanced contributions throughout the dynamics.

**Lemma 1.** *For all $j \in [\![1,p]\!]$, for all $t \geq 0$, $|a_j(t)|^2 - ||w_j(t)||^2 = |a_j(0)|^2 - ||w_j(0)||^2$ , and thus, if $|a_j(0)| \geq ||w_j(0)||$, then $a_j(t)$ maintains its sign and $|a_j(t)| \geq ||w_j(t)||$.*

**Initialization.** Throughout the paper, we initialize the network's weights $w_j$ and $a_j$ from a joint distribution where both marginals are non-zero, centered, rotational-invariant, are sub-Gaussian, and we take the norms of $a_j$ and $w_j$ independent of $d, n, p$. Each pair of neuron is sampled independently from the other pairs. Moreover, we need an assumption of asymmetry of the norm at initialization.

**Assumption 1** (Asymmetric norm at initialization). *We assume that the weights of the network at initialization satisfy for all $j \in [\![1,p]\!]$, $|a_j(0)| \geq ||w_j(0)||$.*

Articles by Boursier and Flammarion [2024a,b] already used this assumption to study two-layer neural networks in order to use the property described in Lemma 1.

**Data.** We define the data matrix $X = (x_1, \ldots, x_n) \in \mathbb{R}^{d \times n}$. Denote $C_x^- = \min_i ||x_i||$ and $C_y^- = \min_i |y_i|$; in what follows, we suppose that $C_x^- > 0$ and $C_y^- > 0$, i.e., the input and output data are bounded away from the origin. Similarly, we also let $C_x^+ = \max_i ||x_i||$ and $C_y^+ = \max_i |y_i|$. We note $C_{x,y}^{+,-}$ to refer to the set of these constants. Finally, we introduce the following hypothesis on the low correlation between the inputs.

**Assumption 2** (Low correlated inputs). *We assume that the data satisfy*

$$||X^T X - D_X|| < \frac{(C_x^-)^2}{2\sqrt{n}} \frac{C_y^-}{C_y^+} , \tag{4}$$

*where $D_X$ denotes the diagonal matrix with coefficients $||x_i||^2$.*

The term $||X^T X - D_X||$ is a control on the magnitude of the correlations $(\langle x_i, x_j \rangle)_{i \neq j}$. As an extreme case, when it equals zero, the inputs are orthogonal. This assumption is purely deterministic at this stage. Later, we show that this weak interaction between the inputs is highly likely to occur for random whitened vectors in high dimensions (see Corollary 1).

**Dimensions.** Throughout the paper, even if the results provided are all non-asymptotic in nature, the reader can picture that the numbers $n, p, d$ (respectively data, neurons and dimension) are all large. Moreover, they verify the following constraint: $n$ is less than $d$, and $p$ can be thought of the order $\log(n)$, meaning only a "low" number of neurons is required.

## 2.1 Related works

**Convergence of neural networks.** Neural networks are known to converge under specific data, parameter, or initialization hypotheses, among which: the neural tangent kernel regime studied by Jacot et al. [2018], Arora et al. [2019], Du et al. [2019], Allen-Zhu et al. [2019], that has been shown to correspond in fact to a *lazy regime* where there is no feature learning because of the initialization scale. Another field of study is the *mean-field* regime, where feature learning can happen but where the optimization has been shown to converge only in the infinite width case [Mei et al., 2018, Chizat and Bach, 2018, Rotskoff and Vanden-Eijnden, 2018]. Note that it is also possible to produce generic counter examples, where convergence does not occur [Boursier and Flammarion,

---

[4]Indeed, the subspace built from all parameters $\theta_\gamma = (\frac{a_j}{\gamma_j}, \gamma_j w_j)_{1 \leq j \leq p}$, when $\gamma$ varies in $(\mathbb{R}_+^*)^p$, maps to the same network, i.e., $h_{\theta_\gamma} = h_{\theta_1}$.

2024b]. Beyond these, there have been attempts to generalize convergence results under local PL (or local curvature) conditions as shown by Chatterjee [2022], Liu et al. [2022], Zhou et al. [2021], but they remain unsatisfactory to explain the good general behavior of neural networks due to the constraint it imposes on the initialization. Convergence theorems similar in spirit to Theorem 1 can be found in an article by Chen et al. [2022]. The main difference relies on two features: only the inner weights are trained and their result necessitates a large value of outer weights when $n$ is large, which is the regime of interest of the present article. Finally, it is worth mentioning other works on neural networks dynamics, e.g., the study of the implicit bias either for regression [Boursier et al., 2022] or classification [Lyu and Li, 2020, Ji and Telgarsky, 2020], or sample complexity to learn functions in a specific context [Glasgow, 2023].

**Polyak-Łojasiewicz properties.** Dating back from the early sixties, Polyak derived a sufficient criterion for a smooth gradient descent to converge to a global minimizer [Polyak, 1964]. This corresponds to the later-called Polyak-Łojasiewicz (PL) constant $\mu$ of a function $f : \mathbb{R}^d \to \mathbb{R}_+$, that can be defined as the best exponential rate of convergence of gradient flow over all initializations, or equivalently to the following minimum ratio $\mu = \min_{x \in \mathbb{R}^d} \frac{\|\nabla f(x)\|^2}{f(x)}$. This has found many applications in non-convex optimization, as it is the case for neural network optimization, and is very popular for optimization in the space of measures [Gentil, 2020]. Other notions of PL conditions have emerged in the literature to characterize local convergence, by bounding the PL constant over a ball $\mu^*(z, r) = \min_{x \in \mathcal{B}(z,r)} \frac{\|\nabla f(x)\|^2}{f(x)}$ [Chatterjee, 2022, Liu et al., 2022] and comparing it to $f(z)$. We use a notion of PL which is local and trajectory-wise to prove lower bounds valid on each trajectory.

## 3   Convergence in high dimension

In this first section, our goal is to understand when the gradient flow converges toward a global minimizer of the loss. Note that the parametrization of the prediction function $h_\theta$ by a neural network often implies the non-convexity of the objective $L$ and prevents any direct application of convex tools in order to ensure global convergence. Generally speaking, even if gradient flows are expected to converge to critical points of the parameter space [Lee et al., 2016], such that $\nabla_\theta L(\theta) = 0$, they might become stuck in local minimizers that do not interpolate the data.

### 3.1   Local PL-curvature

Convexity is not the only tool that provides global convergence: as known in the optimization community, showing that $\frac{\|\nabla L(\theta)\|^2}{L(\theta)}$ is uniformly lower bounded suffices. As mentioned in Section 2.1, this is known as the Polyak-Lojasiewicz condition [Polyak, 1964]. Taking a dynamical perspective on this, we define a trajectory-wise notion of this "curvature" condition which we name the **local-PL curvature** of the system, and define for all $t \geq 0$,

$$\mu(t) := p \frac{\|\nabla L(\theta_t)\|^2}{L(\theta_t)} = -\frac{\frac{d}{dt}L(\theta_t)}{L(\theta_t)} \tag{5}$$

with the second equality being a property of the gradient flow. Intuitively, this coefficient describes the curvature in parameter space that $\theta_t$ "sees" at time $t \geq 0$. The following lemma is classical and shows how it can be used to prove the global convergence of the system, as well as a quantification on the rate.

**Lemma 2.** *Let $\langle \mu(t) \rangle := \frac{1}{t} \int_0^t \mu(u)du$ the time average of the local-PL curvature, which we name the **average-PL curvature**. We have $L(\theta_t) = L(\theta(0))e^{-\langle \mu(t) \rangle t}$.*

Hence, if the **total average-PL curvature** $\langle \mu_\infty \rangle := \lim_{t \to \infty} \langle \mu(t) \rangle$ is strictly positive, we can deduce an upper bound on the loss and convergence to 0 at the exponential speed $\langle \mu_\infty \rangle$. This shows that the average-PL curvature is actually the instantaneous exponential decay rate of the loss, and thus controls the speed at which the system converges.

### 3.2   Global convergence of neural networks for weakly correlated inputs

We are ready to state the main theorem of the paper on the minimization of the loss.

**Theorem 1.** *Let $\varepsilon > 0$, $p \geq 4\log\left(\frac{4n}{\varepsilon}\right)\left(1 + \left(C_{a,w}\frac{C_x^+}{C_y^+}\right)^2\right)$ where $C_{a,w}$ depends only on the joint law of $a, w$, and suppose Assumption 1. We fix the data $(x_i, y_i)_{1 \leq i \leq n}$ and suppose it satisfies Assumption 2. Then with probability at least $1 - \varepsilon$ over the initialization of the network, the loss converges to 0 with $\langle \mu_\infty \rangle \geq \frac{C}{n}$, where we define $C = \frac{6}{5}\frac{(C_x^-)^2}{C_x^+}C_y^-$. Moreover, for any $t \geq 0$, we have the lower bound*

$$\mu(t) \geq \frac{C}{n}\min_i\left|1 - \frac{r_i(t)}{y_i}\right|. \tag{6}$$

Note that, at best, the number of neurons required in Theorem 1 is logarithmic. This finiteness stands in contrast with the infinite number required in the *mean-field regime*, and the polynomial dependency typical of the neural tangent kernel (NTK) regime [Jacot et al., 2018, Allen-Zhu et al., 2019]. In the orthogonal case, the ReLU makes the $\log(n)$ dependency necessary and sufficient, as shown in Lemma 5, as the residual $r_i$ goes to zero if and only if a neuron gets initialized as $a_j y_i > 0$ and $\langle w_j | x_i \rangle > 0$ for each $i$.

Assumption 2 is crucial for this proof: it means that the examples are insufficiently correlated with each other for the weights to collapse onto a single direction. As proved by Boursier and Flammarion [2024a, Theorem 1], the direction $\bar{w}^* = \arg\min_{\theta=\{\bar{w},a\}} L(\theta)$ will attract all neurons if it is accessible from anywhere on the initialization landscape[5]. This phenomenon known as *early alignment* and first described by Maennel et al. [2018], will prevent interpolation if examples are highly correlated [Boursier and Flammarion, 2024a, Theorem 2]. The fact that our result holds for any initialization scale shows that near-orthogonal inputs prevent accessibility to $\bar{w}^*$ and make the early alignment phenomenon benign, as found by Boursier et al. [2022], Frei et al. [2023].

Note finally that our norm-asymmetric initialization (Assumption 1) is sufficient for global convergence with high probability, but may not be necessary. That said, we present in Appendix C.1 a detailed example of a low probability interpolation failure when the assumption is not satisfied.

**Convergence in high dimension.** In this paragraph we assume that the data $(x_i, y_i)_{i=1:n}$ are generated i.i.d. from some distribution $\mathcal{P}_{X,Y}$. We first show that, with high probability, Assumption 2 is almost always valid if the dimension is larger than the square root of the number of data points. Additionally, we assume that the law anti-concentrates at the origin. These two features are gathered in the following lemma.

**Lemma 3.** *Let $(x_i, y_i)_{1 \leq i \leq n}$ be generated i.i.d. from a probability distribution $\mathcal{P}_{X,Y}$ which has compact support on $\mathbb{R}^{*} \times \mathbb{R}^*$, and such that the marginal $\mathcal{P}_X$ has zero-mean, and satisfies $\mathbb{E}_{x \sim \mathcal{P}_X}[xx^T] = \frac{\lambda}{d}I_d$. There exists $C > 0$ depending only on the constants $C_{x,y}^{+,-}$ and the initialization weights, such that, if $d \geq C\left(n^2 + n\log\left(\frac{1}{\varepsilon}\right)\right)$, then, with probability $1 - \varepsilon$, Assumption 2 is satisfied.*

The hypothesis in the previous lemma is satisfied by standard distributions like Gaussians $\mathcal{N}(0, \frac{1}{d}I_d)$ for the inputs. The following corollary restates Theorem 1 for data that are generically distributed as in Lemma 3, and when the dimension is large enough.

**Corollary 1.** *Let $\varepsilon > 0$. Suppose Assumption 1 and that $(x_i, y_i)_{1 \leq i \leq n}$ are i.i.d. generated from a probability distribution satisfying the same properties as in Lemma 3. There exists a constant $C > 0$ depending only on the constants $C_{x,y}^{+,-}$ such that, if the network has $p \geq 4\log\left(\frac{6n}{\varepsilon}\right)\left(1 + \left(C_{a,w}\frac{C_x^+}{C_y^+}\right)^2\right)$ neurons in dimension $d \geq C\left(n^2 + n\log\left(\frac{1}{\varepsilon}\right)\right)$ with $C_{a,w}$ depending only on the join law of $a, w$, then, with probability at least $1 - \varepsilon$ over the initialization of the network and the data generation, the loss converges to 0 at exponential speed of rate at least $\frac{1}{n}$.*

Beyond the high-dimensionality of the inputs, Corollary 1 does not require any initialization specificity (small or large), and the number of neurons required to converge can be as low as $\log(n)$. Hence, let us put emphasis on the fact that the global nice structure of the loss landscape comes from the high-dimensionality: this does not come from a specific region in which the network is

---

[5]This accessibility condition is in fact the absence of saddle point for some function of normed neurons, which imply that neurons can rotate from anywhere on the sphere to $\bar{w}^*$.

initialized as in the NTK (or lazy) regime [Chatterjee, 2022], nor rely on the infinite number of neurons [Wojtowytsch, 2020].

Remark that, under the near-orthogonality assumption, in the large $d$ limit, the largest amount of data that "fits" in the vector space is only $d$, and corresponds to a perturbation of the canonical basis. On average, Corollary 1 finally states that the average number of data points for which we can show convergence is of the order $\sqrt{d}$. Trying to push back this limit up to order $d$ is an important question for future research and seems to ask for other techniques. Experiments underlying this question are presented in Section 5 (Figure 3).

### 3.3 Sketch of Proof

The proof of convergence relies on three key points: **(i)** the loss strictly decreases as long as each example is activated by at least a neuron, **(ii)** for a data point, if there exists a neuron which is activated at initialization, then at least one neuron remains activated throughout the dynamics, **(iii)** At initialization, condition (ii) is satisfied with large probability. Let us detail shortly how each item articulates with one another.

**(i).** First, Lemma 7, stated and proved in Appendix, shows that, by computing the derivatives of the loss, we get a lower bound on the curvature

$$\mu(t) \geq \frac{2}{n}((C_x^-)^2 - ||X^T X - D_X||) \min_i \left\{ \frac{1}{p} \sum_{j=1}^p |a_j|^2 1_{j,i} \right\}. \tag{7}$$

To prove the strict positivity, one needs to show that $||X^T X - D_X||$ is small enough, and that for each data $i$, there exists $j$ such that $|a_j|^2 1_{j,i}$ is strictly positive. Thanks to the initialization of the weights, $|a_j|^2 \geq |a_j(0)|^2 - ||w_j(0)||^2 > 0$, and to Assumption 2, $\frac{1}{2\sqrt{2}}(C_x^-)^2 > ||X^T X - D_X||$. Thus, we have convergence if at any time, for any data input, one neuron remains active, i.e., formally, for all $t \geq 0$, and all $i \in [\![1, n]\!]$, there exists $j \in [\![1, p]\!]$ such that $\langle w_j(t)|x_i \rangle_+ > 0$. Hence, the loss decreases as long as one neuron remains active per data input. We see next how to show this crucial property.

**(ii).** Let us fix the data index $i \in [\![1, n]\!]$, and $y_i > 0$ without loss of generality. Let us define $j_i^* = \arg \max_{a_j y_i > 0} \langle w_j(t)|x_i \rangle$ the index of the largest correctly initialized neuron. Since $a_j$ cannot change sign thanks to Assumption 1, $\langle w_{j_i^*}(t)|x_i \rangle$ is continuous, and has a derivative over each constant segment of $j_i^*$. The strict positivity of this neuron is an invariant of the dynamics: if $r_i \geq y_i$, the derivative of the neuron shows it increases, and if $r_i < y_i$, the residual has decreased, which implies that the $\langle w_{j_i^*}(t)|x_i \rangle$ is strictly positive. Thus, if a neuron is correctly initialized for the data point $i$, a neuron stays active throughout the dynamics. This invariant however requires a large but constant of $n$ number of neurons.

**(iii).** Finally, Lemma 5 shows $\mathbb{P}(\forall i, \exists j, \langle w_j(0)|x_i \rangle > 0 \cap a_j y_i > 0) \geq 1 - n \left(\frac{3}{4}\right)^p$, which implies that for $p \geq 4 \log(\frac{n}{\varepsilon})$, the network is well initialized with probability at least $1 - \varepsilon$.

## 4  Orthogonal Data

In this section, we go deeper on the study of the gradient flow, assuming that the input data are perfectly orthogonal, or equivalently that $||X^T X - D_X|| = 0$. Since most of the intuition for the convergence is drawn from the orthogonal case, it offers stronger results which we detail. In particular, we are able to closely understand the local-PL curvature $(\mu(t))_{t \geq 0}$ evolution and asymptotic behaviour.

### 4.1 Asymptotic PL curvature

Theorem 1 has shown that the local-PL curvature is lower bounded by a term of order $\frac{1}{n}$, allowing us to show an exponential convergence rate of this order. The following proposition shows that in the orthogonal case the curvature can also be upper bounded.

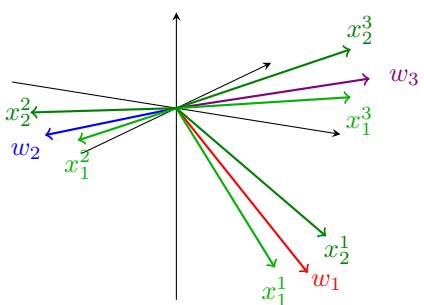

Figure 1: Example of group initialization with $p_n = 3$ neurons, $k_n = 2$ examples per neurons, for $n = 6$ total examples. Group initialization allows to treat each group independently from the other, an thus to solve the problem for a 1-neuron network.

**Proposition 1.** *Let $\varepsilon > 0$. Given orthogonal inputs, and a constant $\Delta > 0$ such that for all $j \in [\![1, p]\!]$, $|a_j(0)| - ||w_j(0)||^2 > \Delta$, there exists $C > 0$ depending only the constants $C_{x,y}^{+;-}$, $\Delta$, and on the law of $a, w$, such that for $d \geq C \log(p) \log(np) \log\left(\frac{1}{\varepsilon}\right)^2$, with probability $1 - \varepsilon$ on the initialization of the network, we have an upper-bound on the local-PL curvature for all $t \geq Cn$,*

$$\mu(t) \leq C\sqrt{\frac{p}{n}} \max_i \left| 1 - \frac{r_i(t)}{y_i} \right| + \frac{C}{n}. \tag{8}$$

This upper bound uses two properties that are characteristic of the orthogonal case. First, once a neuron is inactive on some data input, then, it can never re-activate again. The second property is that for an initialization scale independent on $n$, there is a phase during which correctly initialized neurons increase while the others decrease to 0. This *extinction phase*, proved in Lemma 8, is short in comparison to the time needed to fit the residuals, and leaves the system decoupled between positive and negative outputs $y_i$.

In the limit where $n$ goes to infinity, Proposition 1 shows that the network does not learn since the local-PL is 0. This is an artifact of the orthogonality of the inputs: the interaction between inputs should accelerate the dynamics. However, although all quantities have well defined limits as $n \to +\infty$, the limits cannot be understood as a gradient descent in an infinite dimensional space[6].

Proposition 1 is in fact valid for $p$ fixed, and an initialization of the weights for which every data is correctly initialized by a neuron. In that case, Proposition 1 shows that the asymptotic curvature cannot be larger than the order $\frac{1}{\sqrt{n}}$. While the local-PL curvature is between the order $\frac{1}{n}$ and $\frac{1}{\sqrt{n}}$, the next proposition shows that any intermediate order $\frac{1}{n^\alpha}$, for $\alpha \in [\frac{1}{2}, 1]$, can be reached asymptotically, with strictly positive probability, using a particular initialization of the network.

**Group initialization.**    In the following, we use $p_n$ to denote the number of neurons, and partition the $n$ data points in $p_n$ groups of cardinality $k_n$ (note that $p_n k_n = n$). We re-index the examples per group as by $(x_i^j, y_i^j) = (x_{i+(j-1)k_n}, y_{i+(j-1)k_n})$, for all $i \in [\![1, k_n]\!]$ and $j \in [\![1, p_n]\!]$. Moreover, we use a special initialization of the network such that for all $j, q \in [\![1, p_n]\!]$, $i \in [\![1, k_n]\!]$,

$$\begin{cases} \langle w_j | x_i^q \rangle > 0 \text{ if } j = q \\ \langle w_j | x_i^q \rangle \leq 0 \text{ if } j \neq q \end{cases} \text{ and } \quad a_j = s_j ||w_j||, \tag{9}$$

i.e., $w_j$ is correctly activated on the group $j$ only. An example of group initialization is visible on Figure 1.

**Proposition 2.** *Suppose the group initialization described above, with orthonormal inputs, and the signs of all outputs of the group $j$ are equal to $s_j$. Suppose moreover that the initialization is symmetric, i.e. $|a_j(0)| = ||w_j(0)||$. We fix $k_n = n^{2(1-\alpha)}$ with $\alpha \in [\frac{1}{2}, 1]$. Then, for $t \geq Cn^{3\alpha-1} \log(Cn)$, the local-PL curvature satisfies*

$$\frac{K_1}{n^\alpha} \leq \mu(t) \leq \frac{K_2}{n^\alpha}, \tag{10}$$

---

[6]One would like to write the loss as an expectation over the data point, yet it is impossible as there is no uniform distribution on $\mathbb{N}$.

*where $C = \max\left(\alpha C_y^-, (\frac{1}{2C_y^-})^{\frac{1}{\alpha}}\right)$, $K_1 = 2C_y^- \min_j \frac{||w_j(0)||^2}{2+||w_j(0)||^2}$ and $K_2 = 4C_y^+$.*

Proposition 2 states that any asymptotic value $\langle \mu_\infty \rangle \in [\frac{K_1}{n}, \frac{K_2}{\sqrt{n}}]$ can be achieved with strictly positive probability using group initialization. But of what order is the most likely limit of the curvature for standard initialization? The experiment in Section 5.2 suggest that, with high probability, the asymptotic curvature is always of the order $\frac{1}{\sqrt{n}}$.

**Conjecture 1.** *Let $\varepsilon > 0$. There exist $C_1, C_2 > 0$ depending only on the data and the initialization, such that for $p \geq C_1 \log(\frac{n}{\varepsilon})$ and for orthogonal examples, with probability at least $1 - \varepsilon$ over the initialization of the network, we have convergence of the loss to 0 and*

$$\langle \mu_\infty \rangle = \frac{C_2}{\sqrt{n}}. \tag{11}$$

### 4.2 Phase transition in the PL curvature

In the previous section, we emphasized the asymptotic order of the local-PL curvature with respect to $n$ and hypothesized that it is of the order $\frac{1}{\sqrt{n}}$ in most cases. In this section, we are interested in the evolution of the local-PL curvature during the dynamics. Lemma 4 below computes the local-PL curvature at initialization in the large $p$ regime, and shows that initially it is of order $\frac{1}{n}$.

**Lemma 4.** *At initialization, the local-PL curvature $\mu(0)$ is a random variable which satisfy $\sqrt{p}(n\mu(0) - \beta_0) \underset{p \to +\infty}{\longrightarrow} \mathcal{N}\left(0, \gamma_0^2\right)$, and with $\beta_0, \gamma_0$ depending only on the data and the distributions of the network's neurons.*

The constant $\beta_0$ is strictly positive as soon as the limit network does not directly equal the labels, which is natural to assume since they are unknown a priori. Thus the exponential rate of decrease of the loss in the early times of the dynamics is of order $\frac{1}{n}$. Importantly, Proposition 2 with a single group has an asymptotic speed of order $\frac{1}{\sqrt{n}}$, meaning that the local-PL curvature transitions between $\frac{1}{n}$ and $\frac{1}{\sqrt{n}}$. If Conjecture 1 is true, then this phenomenon happens with high probability during the dynamics.

Let us study this phenomenon through the example of Proposition 2, with a fixed number of neurons $p$. In this case, the following theorem shows that there are exactly $p$ phase transitions of the loss, which each corresponds to a data group being fitted. To be precise, let us define $L_\infty(t) = \lim_{n \to +\infty} L_n(t)$, with $L_n(t) = L(\theta(t \times t_n))$, $t_n = \frac{\sqrt{np}}{4} \log(np)$, and $p$ fixed ($k_n = \frac{n}{p}$). We prove that $L_\infty$ is constant by parts with at most $p$ parts.

**Theorem 2.** *Suppose the same data hypothesis and initialization as Proposition 2. We define $||D_j^n||^2 = \frac{1}{k_n} \sum_{i=1}^{k_n} (y_i^j)^2$ for each cluster, and suppose its limit $||D_j^\infty||^2$ finite. Then, the function $L_\infty$ is constant by parts with at most $p$ parts, and the transitions happen at each time $t^j = \frac{1}{||D_j^\infty||}$. Moreover, for all $\varepsilon \in ]0, 1[$, there exist times $t_n^j(\varepsilon)$ satisfying*

$$L^j(t_n^j(\varepsilon)) = \frac{\varepsilon}{2}||D_j^n||^2, \quad \frac{t_n^j(\varepsilon)}{t_n^j(1-\varepsilon)} \sim_n 1 \text{ and } \frac{t_n^j(1-\varepsilon) - t_n^j(\varepsilon)}{t_n} \sim_n \frac{1}{2||D_j^\infty||} \frac{\log\left(C^j(\varepsilon)\right)}{\log(n)}, \tag{12}$$

*where $L^j$ is the part of the loss corresponding to the group $j$, and $C^j(\varepsilon) > 1$ depends on $\varepsilon$ and the initializations and data of the group $j$.*

The theorem shows that each transition of $L_n$ occurs in the time frame which decreases as $\frac{1}{\log(n)}$. Note that these transitions are subtle: one needs extremely large dimensions in order to differentiate two close transitions as shown on Figure 2. The phase transitions of the loss are in fact associated with transitions of $||w_j||^2$ from a constant order to an order $\sqrt{n}$, and by Lemma 7 with transitions on the local-PL from order $\frac{1}{n}$ to order $\frac{1}{\sqrt{n}}$.

## 5 Experiments

In this Section, we aim to perform deeper experimental investigations on the system, which we could not do formally. Precisely, we want to answer two questions:

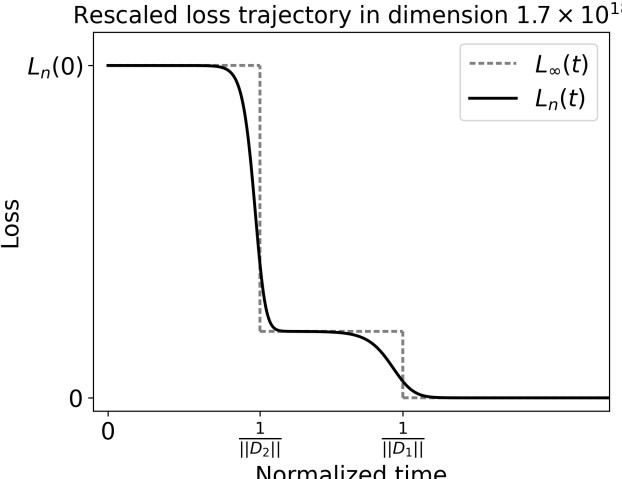

Figure 2: Simulation of the loss trajectory of a network with 2 neurons and group initialization, each activated separately on half the data points. $L_n$ is the rescaled loss for $n$ examples, and $L_\infty$ is its limit as $n$ goes to infinity. We can see two phase transitions in the very high-dimensional regime.

1. What is the probability that the loss reaches 0 for $n$ data points in dimension $d$, under the distributional hypotheses of Lemma 3 (sub-Gaussian, zero-mean and whitened data)? What is the maximum $n$ for a fixed $d$ such that global convergence holds with high probability ?

2. In the orthogonal case, is the asymptotic exponential convergence rate of order $\frac{1}{\sqrt{n}}$ (on average over the initialization) as stated in Conjecture 1?

The data and weights distribution which have been used for the experiments below can be found in Appendix B, and the code is available on GitHub.

## 5.1 Probability of Convergence

This section aims to test the limit in which Corollary 1 holds when the number of data points increase. Intuitively, as the number of examples $n$ grows, the neural network becomes less and less overparametrized, and hence is expected to fail to globally converge. Knowing if and when this occurs with high probability is important for us to understand how much our current threshold $C\sqrt{d}$ can be improved. We thus plot the probability of convergence, as well as the loss at convergence to obtain additional information when the probability is zero. We train 500 one-layer neural networks with the normalization presented in Section 2, dimension $d = 100$, $n$ ranging from 2500 to 3500, and $p_n = C \log(n)$ neurons. Additional details on the training procedure can be found in Appendix B.

Figure 3 shows that for $n \leq 2900$, the probability of convergence is very likely, for $n \geq 3100$ the probability is almost zero, and in between, there is a sharp transition. This sharp transition is visible for any value $d$ at some point $N(d, p)$, which we name the **convergence threshold**. By measuring the point for different values of $d$ and $p$, we see that the threshold scales like $N(d, p) \simeq C(p)d$, with $C(p)$ which is sub-linear (see Figure 5 in Appendix B). In particular, for $n \leq Cdp$, there exists a network that interpolates the data, meaning that the convergence threshold is not a threshold for the existence of a global minimum. The threshold's scaling is linear in $d$ which implies that proving convergence for $Cd$ data in dimension $d$ seems feasible.

## 5.2 Empirical asymptotic local-PL curvature

In this section we test Conjecture 1, and to do so we measure $\mu(t)$ during the dynamics, and mostly at the end of the dynamics, since we know by Lemma 4 that near 0 the local-PL curvature is of order $\frac{1}{n}$. To provide the strongest evidence for the conjecture, we measured the order of the local-

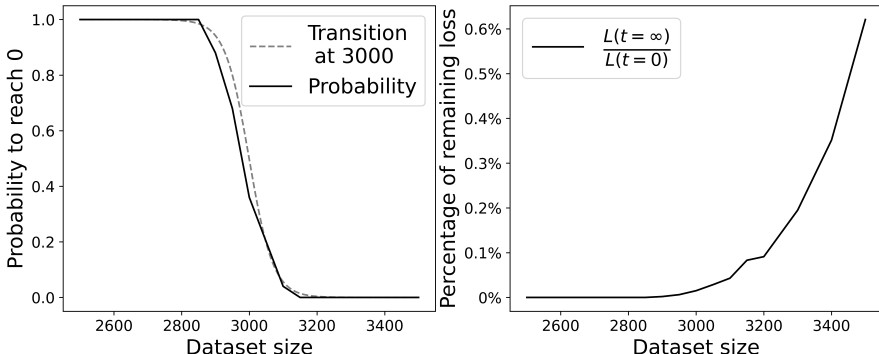

Figure 3: **Left:** Probability that a network trained on $n$ data converges to 0 loss. We observe a transition at $n = 3000$, from likely to unlikely convergence.
**Right:** Loss at convergence normalized by the loss at initialization. For $n \geq 3000$, the loss increases to $0.6\%$, which is equivalent to fitting all but one example.

PL curvature in three ways: by directly measuring the local-PL $\mu(t_\infty) = \log\left(\frac{L(t_\infty - 1)}{L(t_\infty)}\right)$ at the last epoch $t_\infty$, by measuring the average-PL curvature $\langle \mu_\infty \rangle = \frac{1}{t_\infty} \log\left(\frac{L(0)}{L(t_\infty)}\right)$, and finally by mesuring the lower and upper bounds on the local-PL given in Lemma 7.

Following Conjecture 1, all approximations should likely be decreasing in $\frac{1}{\sqrt{n}}$ as $n$ increases. To show this, we plot the log-log graph of each measure above. We train 500 networks in dimension $d = 2000$, with $n$ ranging from 1000 to 2000, and $p_n = C \log(n)$. All resulting plots appear linear in the log-log scale, with a slope close to $-\frac{1}{2}$ (see Figure 6 in Appendix B), meaning that the scalings are indeed in $\frac{C}{\sqrt{n}}$. This empirically confirms our conjecture that the local-PL curvature has order $\frac{1}{\sqrt{n}}$ asymptotically.

# 6 Conclusion

We have studied the convergence of the gradient flow on one-hidden-layer ReLU networks with finite datasets. Our analysis leverages a local Polyak-Łojasiewicz viewpoint on the gradient-flow dynamics, revealing that for a large dimension $d$ in the order of $n^2$ data points, we can **guarantee global convergence with high probability** using only $\log(n)$ neurons. The specificity of the system relies on the low-correlation between the input data due to the high dimension. Moreover, in the orthogonal setting the **loss's exponential rate of convergence** is at least of order $\frac{1}{n}$ and at most of order $\frac{1}{\sqrt{n}}$, which is also the average asymptotic order as experimentally verified. For a special initialization of the network, a **phase transition in this rate** occurs during the dynamics.

**Future Directions.** We are most enthusiastic about proving the convergence of the networks for linear threshold $d \geq Cn$, which should require new proof techniques, as well as quantifying the impact of large amounts of neurons on the system, which has been overlooked in our study. Future work should also consider using a teacher-network to generate the outputs, in order to link the probability or interpolation with the complexity, in terms of neurons, of the teacher.

# Acknowledgements

This work has received support from the French government, managed by the National Research Agency, under the France 2030 program with the reference "PR[AI]RIE-PSAI" (ANR-23-IACL-0008).

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

## ORGANIZATION OF THE APPENDIX

The appendices of this article are structured as follows. Appendix A contains the proofs of each of the 10 statements of the paper in an entitled subsection, with additional lemmas included in the relevant subsections. Only Corollary 1 doesn't have a complete proof as it is a simple combination of Lemma 3 and Theorem 1. Appendix B contains additional details on the experiments that were performed in Section 5, as well as graphs for the scaling laws. Finally, Appendix C contains general discussions about the possibility to provably learn $2d$ inputs in dimension $d$, and the possible collapse of the second-layer weights.

## A   Proofs

Let us note a few equations that we will use as references for the proofs below. First, the two equations from the gradient descent in (3) are the following.

$$\frac{d}{dt}w_j = \frac{a_j}{n}\sum_{i=1}^{n} r_i x_i 1_{j,i} = \frac{a_j}{n}XP_jR \tag{13}$$

$$\frac{d}{dt}a_j = \frac{1}{n}\sum_{i=1}^{n} r_i \langle w_j|x_i\rangle_+ = \frac{1}{n}w_j^T XP_jR \tag{14}$$

In matrix notation, $R(t)$ is the column vector of the residuals, $X$ is the data matrix, and $P_j$ is the diagonal matrix with diagonal elements $1_{i,j} = \mathbb{1}_{\langle w_j|x_i\rangle_+>0}$. A second important derivative of the system is then the residuals.

$$\begin{aligned}
\frac{d}{dt}r_i &= -\frac{1}{p}\sum_{j=1}^{p}\left(\frac{d}{dt}a_j\right)\langle w_j|x_i\rangle_+ + a_j\left(\frac{d}{dt}\langle w_j|x_i\rangle_+\right) \\
&= -\frac{1}{np}\sum_{j=1}^{p} 1_{j,i}x_i^T w_j w_j^T XP_jR + |a_j|^2 1_{j,i}x_i^T XP_jR \\
\frac{d}{dt}R &= -\frac{1}{np}\left[\sum_{j=1}^{p} P_j X^T w_j w_j^T XP_j + |a_j|^2 P_j X^T XP_j\right]R \\
&= -\frac{1}{n}MR
\end{aligned} \tag{15}$$

with $M$ a time dependent symmetric matrix. Finally, taking the product with $R$ in equation (15), we obtain an equation on the local-PL curvature.

$$\begin{aligned}
\frac{d}{dt}L(t) &= -\frac{2}{n}R^T(t)M(t)R(t) \\
\mu(t) &= \frac{2}{n}\bar{R}^T(t)M(t)\bar{R}(t)
\end{aligned} \tag{16}$$

where we recall that $\bar{R} = \frac{R}{||R||}$.

### A.1   Theorem 1

Lemma 5 shows that a number of neurons of order $\log(n)$ is both necessary and sufficient to obtain the event $\mathcal{I}$, which corresponds to an initialization of the network which guarantees convergence.

**Lemma 5.** *Suppose $y_i \neq 0$, and let $\mathcal{I}$ be the event: for all $i$, there exists $j$ such that, $\langle w_j(0)|x_i\rangle > 0$ and $a_j(0)y_i > 0$. For all $\varepsilon > 0$,*

- *if $p \geq 4\log(\frac{n}{\varepsilon})$, then $\mathbb{P}(\mathcal{I}) \geq 1 - \varepsilon$,*

- *if $p \leq 3\log(\frac{n}{\varepsilon}) - 2$, then $\mathbb{P}(\mathcal{I}) \leq 1 - \varepsilon$,*

*and thus, $\mathbb{P}(\mathcal{I}) = 1 - \varepsilon$ implies $p \in [\![3\log(\frac{n}{\varepsilon}) - 2, 4\log(\frac{n}{\varepsilon})]\!]$.*

*Proof.* **of Lemma** 5

Let us note $\langle w_j(0)|x_i\rangle = W_{i,j}$ and $A_j = a_j(0)y_i$ random variables which are symmetric. $A_j$ are independent with all variables, while $W_{j,i}$ are independent with all variables $A_j$ and $W_{q,k}$ with $q \neq j$.

$$\mathbb{P}(\mathcal{I}) = \mathbb{P}(\forall i, \exists j, \langle w_j(0)|x_i\rangle > 0 \cap a_j y_i > 0)$$

$$= \mathbb{P}\left(\bigcap_i \bigcup_j W_{i,j} > 0 \cap A_j > 0\right)$$

$$= 1 - \mathbb{P}\left(\bigcup_i \bigcap_j W_{i,j} \leq 0 \cup A_j \leq 0\right)$$

$$\geq 1 - n\mathbb{P}\left(\bigcap_j W_{i,j} \leq 0 \cup A_j \leq 0\right) \tag{17}$$

$$= 1 - n\mathbb{P}\left(W_{i,j} \leq 0 \cup A_j \leq 0\right)^p$$

$$= 1 - n(1 - \mathbb{P}\left(W_{i,j} > 0 \cap A_j > 0\right))^p$$

$$= 1 - n\left(1 - \mathbb{P}\left(W_{i,j} > 0\right)\mathbb{P}\left(A_j > 0\right)\right)^p$$

$$= 1 - n\left(\frac{3}{4}\right)^p$$

Replacing the expression with $p = 4\log(\frac{n}{\varepsilon}) \geq \frac{1}{\frac{1}{3}-\frac{1}{2}\frac{1}{3^2}}\log(\frac{n}{\varepsilon}) \geq \frac{\log(\frac{n}{\varepsilon})}{\log(\frac{4}{3})}$, we find that the probability is larger than $1 - \varepsilon$. Now for the other bound,

$$\mathbb{P}(\mathcal{I}) = \mathbb{P}(\forall i, \exists j, \langle w_j(0)|x_i\rangle > 0 \cap a_j y_i > 0)$$

$$= \mathbb{P}\left(\bigcap_i \bigcup_j W_{i,j} > 0 \cap A_j > 0\right)$$

$$\leq \mathbb{P}\left(\bigcap_i \bigcup_j W_{i,j} > 0\right)$$

$$= \mathbb{P}\left(\bigcup_j W_{1,j} > 0\right)^n \tag{18}$$

$$= (1 - \mathbb{P}\left(W_{1,1} > 0\right)^p)^n$$

$$= (1 - 2^{-p})^n$$

$$\leq \left(1 - 4\left(\frac{\varepsilon}{n}\right)^{3\log(2)}\right)^n$$

$$\leq \left(1 - 4\frac{\varepsilon}{n}\right)^n$$

$$\leq 1 - \varepsilon$$

where we use $(1 - \frac{x}{n})^n \leq 1 - \frac{x}{e} \leq 1 - \frac{x}{4}$ valid on $x \in [0, 1]$. $\square$

**Lemma 6.** *Let $\varepsilon > 0$, and $p \geq \frac{1}{c}\log\left(\frac{2}{\varepsilon}\right)\max\left(\left(||a\langle w|\bar{x}\rangle_+||_{\psi_1}\frac{C_x^+}{C_y^+}\right)^2, ||a\langle w|\bar{x}\rangle_+||_{\psi_1}\frac{C_x^+}{C_y^+}\right)$ for any vector $x$, $c > 0$ a constant, and $||\cdot||_{\psi_1}$ the sub-exponential norm. We have the following bound on the loss at initialization.*

$$\mathbb{P}\left(L(\theta_0) \leq 2(C_y^+)^2\right) \geq 1 - \varepsilon \tag{19}$$

*Proof.* **of Lemma 6**

First, let us upper bound the loss at $t = 0$.

$$L(\theta_0) = \frac{1}{2n} \sum_{i=1}^{n} r_i^2$$

$$\leq \frac{1}{2n} \sum_{i=1}^{n} \left( y_i - \frac{1}{p} \sum_{j=1}^{p} a_j \langle w_j | x_i \rangle_+ \right)^2 \tag{20}$$

$$\leq \frac{1}{2} \left( y_I - \frac{1}{p} \sum_{j=1}^{p} a_j \langle w_j | x_I \rangle_+ \right)^2$$

Since $a, w$ are sub-Gaussian random variables, the product is a centered sub-exponential random variable. Now, using Theorem 2.9.1 from Vershynin [2018], we get the following bound

$$\mathbb{P}\left( \left| \frac{1}{p} \sum_{j=1}^{p} a_j \langle w_j | x_I \rangle_+ \right| \leq K \right) \geq 1 - \mathbb{P}\left( \left| \frac{1}{p} \sum_{j=1}^{p} a_j \langle w_j | x_I \rangle_+ \right| > K \right)$$

$$\geq 1 - 2e^{-cp \min\left( \frac{K^2}{||a\langle w_j | x_I \rangle_+||_{\psi_1}^2}, \frac{K}{||a\langle w_j | x_I \rangle_+||_{\psi_1}} \right)} \tag{21}$$

where $c$ is an absolute constant. Taking $K = ||a\langle w_j | x_I \rangle_+||_{\psi_1} \max\left( \frac{\log\left(\frac{2}{\varepsilon}\right)}{cp}, \sqrt{\frac{\log\left(\frac{2}{\varepsilon}\right)}{cp}} \right)$, we get that

the sum is bounded with probability at least $1 - \varepsilon$. Now using the inequality on $p$ in the statement and that $w$ as a distribution invariant by rotation, we obtain $K \leq C_y^+$, and thus

$$L(\theta_0) \leq 2(C_y^+)^2. \tag{22}$$

$\square$

**Lemma 7.** *For any set of parameters $\theta = (a_j, w_j)_{j=1:p}$, the following bounds on the local-PL curvature hold.*

$$\mu(t) \leq \frac{2}{n}(C_x^+)^2 \max_i \frac{1}{p} \sum_{j=1}^{p} (|a_j|^2 + ||w_j||^2) 1_{j,i}$$

$$\mu(t) \geq \frac{2}{n}((C_x^-)^2 - ||X^T X - D_X||) \min_i \frac{1}{p} \sum_{j=1}^{p} |a_j|^2 1_{j,i} \tag{23}$$

*where we recall that $D_X$ denotes the diagonal matrix with coefficients $||x_i||^2$.*

*Proof.* **of Lemma 7**

We start from equation (16), which shows that the local-PL curvature lies between the largest and smallest eigen values of the symmetric matrix $M(t)$.

$$\mu(t) = \frac{2}{n} \left[ \frac{1}{p} \sum_{j=1}^{p} (w_j^T X P_j \bar{R})^2 + |a_j|^2 ||X P_j \bar{R}||^2 \right] \tag{24}$$

By the triangular inequality, we have $0 \leq (w_j^T X P_j \bar{R})^2 \leq ||w_j||^2 ||X P_j \bar{R}||^2$, which gets us bounds on the local-PL curvature.

$$\frac{2}{np} \sum_{j=1}^{p} |a_j|^2 ||X P_j \bar{R}||^2 \leq \mu(t) \leq \frac{2}{np} \sum_{j=1}^{p} (||w_j||^2 + |a_j|^2) ||X P_j \bar{R}||^2 \tag{25}$$

We transform the term $||X P_j \bar{R}||^2$ to make $||X^T X - D_X||$ appear.

$$||X P_j \bar{R}||^2 = \bar{R}^T P_j^T X^T X P_j \bar{R}$$

$$= \bar{R}^T P_j^T (D_X - (X^T X - D_X)) P_j \bar{R} \tag{26}$$

$$= ||\sqrt{D_X} P_j \bar{R}||^2 - ||\sqrt{X^T X - D_X} P_j \bar{R}||^2$$

Now, by bounding $\sqrt{D_X}$ by its largest and smallest eigen values, namely $C_x^+$ and $C_x^-$, we get the bounds on this term.

$$(C_x^-)^2||P_j\bar{R}||^2 - ||X^TX - D_X||||P_j\bar{R}||^2 \leq ||XP_j\bar{R}||^2 \leq (C_x^+)^2||P_j\bar{R}||^2 \qquad (27)$$

The next step is to find the lower bound on the two remaining terms.

$$\frac{1}{p}\sum_{j=1}^{p}|a_j|^2||P_j\bar{R}||^2 = \frac{1}{p}\sum_{j=1}^{p}|a_j|^2\frac{||P_jR||^2}{||R||^2} = \frac{1}{p}\sum_{j=1}^{p}|a_j|^2\frac{\sum_{i=1}^{n}r_i(t)^2 1_{i,j}}{\sum_{i=1}^{n}r_i(t)^2} \qquad (28)$$

By inverting the sum in $i$ and $j$, and taking the minimum of maximum over $j$, we get the bounds.

$$\frac{1}{p}\sum_{j=1}^{p}|a_j|^2||P_j\bar{R}||^2 \geq \min_i \frac{1}{p}\sum_{j=1}^{p}|a_j|^2 1_{i,j} \qquad (29)$$

$$\frac{1}{p}\sum_{j=1}^{p}(|a_j|^2 + ||w_j||^2)||P_j\bar{R}||^2 \leq \min_i \frac{1}{p}\sum_{j=1}^{p}(|a_j|^2 + ||w_j||^2)1_{i,j} \qquad (30)$$

From equation (25), we use the equation (27) as well as equation (29) for the lower bound and (30) for the upper bound and find the expected bounds on the local-PL curvature. $\qquad \square$

*Proof.* **of Theorem 1**
This is a proof based on the sketch visible in Section 3. The proof of convergence relies on three key points:

(i) The loss strictly decreases as long as each example is activated by at least a neuron.

(ii) For a data point, if there exists a neuron which is activated at initialization, then at least one neuron remains activated throughout the dynamics.

(iii) At initialization, the previous condition is satisfied with large probability.

We finish the proof with the lower bounds on $\mu(t)$ and $\langle\mu_\infty\rangle$.

**(i)** First, Lemma 7 shows that, by computing the derivatives of the loss, we get a lower bound on the curvature.

$$\mu(t) \geq \frac{2}{n}\left((C_x^-)^2 - ||X^TX - D_X||\right)\min_i\left\{\frac{1}{p}\sum_{j=1}^{p}|a_j|^2 1_{j,i}\right\} \qquad (31)$$

We want to show the strict positivity of this lower bound. First, using Assumption 2, we have for all $n \geq 2$ that

$$(C_x^-)^2 - ||X^TX - D_X|| \geq (C_x^-)^2\left(1 - \frac{1}{2\sqrt{n}}\frac{C_y^-}{C_y^+}\right) \geq \left(1 - \frac{1}{2\sqrt{2}}\right)(C_x^-)^2 \geq \frac{3}{5}(C_x^-)^2 \qquad (32)$$

which also holds for $n = 1$ since then $||X^TX - D_X|| = 0$. Moreover, thanks to the asymmetric initialization, we have $|a_j|^2 \geq |a_j(0)|^2 - ||w_j(0)||^2 > 0$, which means that $\mu(t)$ is bounded away from 0 as long as for all $i$ there exists $j$ satisfying $\langle w_j(t)|x_i\rangle_+ > 0$, *i.e.*, that $1_{i,j} = 1$.

**(ii)** Let us fix the data index $i \in [\![1, n]\!]$, and $y_i > 0$ without loss of generality. Let us define the index of the largest correctly initialized neuron $j_i^*$.

$$j_i^* = \arg\max_{a_j y_i > 0}\langle w_j(t)|x_i\rangle \qquad (33)$$

Since $a_j$ cannot change sign thanks to Assumption 1, $\langle w_{j_i^*}(t)|x_i\rangle$ is continuous, and has a derivative over each constant segment of $j_i^*$. We can write the derivatives of this neuron as

$$
\begin{aligned}
\frac{d}{dt}\langle w_{j_i^*}|x_i\rangle &= \frac{a_{j_i^*}}{n}\sum_k^n r_k\langle x_i|x_k\rangle 1_{j_i^*,k} \\
&= \frac{a_{j_i^*}}{n}e_i^T X^T X P_{j_i^*}R \\
&= \frac{a_{j_i^*}}{n}e_i^T(D_X-(X^T X-D_X)P_{j_i^*}R \\
&\geq \frac{|a_{j_i^*}|}{n}\left[r_i\|x_i\|^2 1_{j_i^*,i}s_{j_i^*}-\|X^T X-D_X\|\|R\|\right] \\
&= \frac{|a_{j_i^*}|}{n}\left[r_i\|x_i\|^2 1_{j_i^*,i}s_{j_i^*}-\|X^T X-D_X\|\sqrt{2nL(\theta_t)}\right] \\
&\geq \frac{|a_{j_i^*}|}{n}\left[r_i\|x_i\|^2 1_{j_i^*,i}s_{j_i^*}-\|X^T X-D_X\|\sqrt{2nL(\theta_0)}\right]
\end{aligned}
\tag{34}
$$

We use Assumption 2 to have $\|X^T X-D_X\| < \frac{1}{2\sqrt{n}}(C_x^-)^2\frac{C_y^-}{C_y^+}$, and lemma 6 with $\frac{\varepsilon}{2}$ to have $L(\theta_0)\leq 2(C_y^+)^2$ with propability at least $1-\frac{\varepsilon}{2}$. Moreover, $s_{j_i^*}y_i > 0$, gets us the following inequality.

$$
\frac{d}{dt}\langle w_{j_i^*}|x_i\rangle > \frac{C_y^-(C_x^-)^2}{n}|a_{j_i^*}|\left[\frac{r_i}{y_i}1_{j_i^*,i}-1\right]
\tag{35}
$$

Now, the strict positivity of $\langle w_{j_i^*}|x_i\rangle$ is an invariant of the system: if $\frac{r_i}{y_i}\geq 1$, then $\langle w_{j_i^*}|x_i\rangle$ strictly increases, and otherwise we have

$$
0 < \frac{y_i-r_i}{y_i}=\frac{1}{p}\sum_{j=1}^p\frac{a_j}{y_i}\langle w_j|x_i\rangle_+ \leq \langle w_{j_i^*(t)}|x_i\rangle\frac{1}{p}\sum_{j=1}^p\frac{|a_j|}{|y_i|}
\tag{36}
$$

Which implies that $\langle w_{j_i^*}|x_i\rangle$ stays strictly positive throughout the dynamics.

**(iii)** As shown in Lemma 5, for $p\geq 4\log\left(\frac{2n}{\varepsilon}\right)$, we have the strict positivity with probability $1-\frac{\varepsilon}{2}$.

$$
\mathbb{P}(\forall i,\exists j,\langle w_j(0)|x_i\rangle > 0\cap a_j y_i > 0)\geq 1-\varepsilon.
\tag{37}
$$

Finally, we prove the lower bounds on the PL. Let us recall that $|a_j|\geq\|w_j\|$ and that $1_{j,i}\geq\langle\bar{w}_j(t)|\bar{x}_i\rangle_+$, which gives us

$$
\frac{1}{p}\sum_{j=1}^p|a_j|^2 1_{j,i}\geq\left|\frac{1}{p}\sum_{j=1}^p a_j\langle w_j|\bar{x}_i\rangle_+\right|=\left|\frac{y_i-r_i(t)}{C_x^+}\right|.
\tag{38}
$$

We can plug these into equation (31) to obtain

$$
\mu(t)\geq\frac{6}{5n}\frac{(C_x^-)^2}{C_x^+}C_y^-\min_i\left|1-\frac{r_i(t)}{y_i}\right|
\tag{39}
$$

We obtain the final lower bound on the local-PL curvature by seeing that $2-\sqrt{2}\geq\frac{1}{2}$. From this last equation, by integration, we obtain

$$
\frac{1}{t}\int_0^t\mu(u)du\geq\frac{6}{5n}\frac{(C_x^-)^2}{C_x^+}C_y^-\left(1-\frac{1}{t}\int_0^t\max_i\left|\frac{r_i(u)}{y_i}\right|du\right)
\tag{40}
$$

Let $t_\delta$ satisfying $\max_i|r_i(t)|\leq\delta$ for all $t\geq t_\delta$. $t_\delta$ exists and is finite since the loss reaches 0. Thus, we have for any $\delta > 0$ that

$$
\frac{1}{t}\int_0^t\mu(u)du\geq\frac{6}{5n}\frac{(C_x^-)^2}{C_x^+}C_y^-\left(1-\frac{t_\delta}{t}\max_i\frac{\sqrt{2nL(\theta_0)}}{|y_i|\sqrt{p}}-\frac{t-t_\delta}{t}\delta\right)
\tag{41}
$$

whic in the limit $t \to +\infty$ gives

$$\langle \mu_\infty \rangle \geq \frac{6}{5n} \frac{(C_x^-)^2}{C_x^+} C_y^- (1 - \delta) \tag{42}$$

Taking $\delta \to 0$ gives the desired bound on the average-PL curvature. In total, we use two bounds valid with probability $1 - \frac{\varepsilon}{2}$, we by the union bound, the Theorem is valid with probability at least $1 - \varepsilon$. Moreover, we check that taking in the statement $C_{a,w} = \frac{1}{x} ||a \langle w | \bar{x} \rangle||_{\psi_1}$, allows us to use both lemmas 5 and 6. $\square$

## A.2 Lemma 3

*Proof.* **of Lemma 3**
This proof heavily relies on the result of Vershynin [2010, Remark 5.59] on the concentration of sub-Gaussian random variables. It states that if $A \in \mathbb{R}^{n \times d}$ is a matrix, the columns of which are $n$ independent centered, whitened[7], sub-Gaussian random variables in dimension $d$, then with probability $1 - 2e^{-t^2}$,

$$\left|\left|\frac{1}{d} A^T A - I_d\right|\right| \leq C\sqrt{\frac{n}{d}} + \frac{t}{\sqrt{d}} \tag{43}$$

with $C > 0$ depending only on $\max_i ||A_i||_{\psi_2}$ the sub-Gaussian norm of the columns. We use this property with $A_i = \sqrt{d} \bar{x}_i$ which satisfies every hypothesis, in particular it is sub-Gaussian since the norm is constant. Taking $t = \sqrt{\log\left(\frac{2}{\varepsilon}\right)}$, we obtain the following bound.

$$||\bar{X}^T \bar{X} - I_d|| = \left|\left|\frac{1}{d} A^T A - I_d\right|\right| \leq C\sqrt{\frac{n}{d}} + \sqrt{\frac{\log\left(\frac{2}{\varepsilon}\right)}{d}} \tag{44}$$

Moreover, we can link this concentration inequality with the control term of Assumption 2.

$$\begin{aligned}
||X^T X - D_X|| &\leq \left|\left| D_X^{\frac{1}{2}} (D_X^{-\frac{1}{2}} X^T X D_X^{-\frac{1}{2}} - I_d) D_X^{\frac{1}{2}} \right|\right| \\
&\leq \left|\left| D_X^{\frac{1}{2}} \right|\right|^2 \left|\left| D_X^{-\frac{1}{2}} X^T X D_X^{-\frac{1}{2}} - I_d \right|\right| \\
&\leq (C_x^+)^2 \left|\left| \bar{X}^T \bar{X} - I_d \right|\right| \\
&\leq (C_x^+)^2 \left( C\sqrt{\frac{n}{d}} + \sqrt{\frac{\log\left(\frac{2}{\varepsilon}\right)}{d}} \right)
\end{aligned} \tag{45}$$

Thus, the condition in Assumption 2 is satisfied with probability as least $1 - \varepsilon$ if

$$d \geq 8 \left[\frac{C_y^+}{C_y^-}\right]^2 \left[\frac{C_x^+}{C_x^-}\right]^4 \left( C^2 n^2 + n \log\left(\frac{2}{\varepsilon}\right) \right). \tag{46}$$

Recall that $C_{x,y}^{+,-}$ are independent of $n$ since $\mathcal{P}_{X,Y}$ has compact support away from 0. $\square$

*Proof.* **of Corollary 1**
To prove the corollary, we simply use Lemma 3 instead of Assumption 2 in the proof of Theorem 1. Moreover, we check that taking $p$ as in the statement allows us to have each Lemma 3, 5, and 6 with $\frac{\varepsilon}{3}$. $\square$

## A.3 Proposition 1

**Lemma 8.** *Let $\varepsilon > 0$. Suppose that*

  1. *the $(x_i)_{i=1:n}$ form an orthogonal family of non zero vectors, and that $(y_i)_{i=1:n}$ are non-zero,*

---

[7]In this article, Vershynin [2010, Remark 5.59] uses the isotropy of the columns, but defines it as $\mathbb{E}[xx^T] = I_d$, which we rather refer to as a whitened distribution.

2. *for all $j \in [\![1, p]\!]$, $|a_j(0)|^2 - ||w_j(0)||^2 \geq \Delta$ for some constant $\Delta > 0$,*

3. *and that for all $i \in [\![1, n]\!]$, there exist $j \in [\![1, p]\!]$ such that $\langle w_j(0)|x_i\rangle > 0$ and $a_j(0)y_i > 0$.*

*Then, there exist a constant $\kappa$ depending only on $C_{x,y}^{+,-}$, $\Delta$, $||a||_{\psi_2}$ and $\left|\left|(||w||_2)^2\right|\right|_{\psi_2}$ such that if $d \geq \kappa \log(p) \log(np) \log\left(\frac{1}{\varepsilon}\right)^2$, then with probability at least $1 - \varepsilon$ on the initialization of the network, at $t_n = \frac{2n}{C_y^- C_x^- \Delta} \max_{j,i}\langle w_j(0)|x_i\rangle_+$, we have*

$$
\begin{aligned}
a_j(0)y_i > 0 &\implies \langle w_j(t_n)|x_i\rangle_+ \geq \langle w_j(0)|x_i\rangle_+ \\
a_j(0)y_i < 0 &\implies \langle w_j(t_n)|x_i\rangle_+ \leq 0.
\end{aligned}
\tag{47}
$$

This Lemma states that, for orthogonal data, incorrectly initialized neuron, *i.e.* neurons for which $a_j y_i < 0$, vanish in finite time, and cannot become active again. Thus, after time $t_n$, the system is decoupled between the positive and negative labels, and only correctly initialized neuron, which are useful to the prediction, persist.

In particular, it is possible to show that neurons vanish if $y_i = 0$, but the vanishing doesn't happen in finite time.

*Proof.* **of Lemma 8**
We start by computing the derivative of a neuron in the orthogonal setting, which is given from equation (13).

$$
\frac{d}{dt}\langle w_j|x_i\rangle_+ = \frac{a_j r_i 1_{j,i} ||x_i||^2}{n}
\tag{48}
$$

This equation shows that if a neuron is null or negative at any point in time, then it stays at 0. Thus, let us only discuss the case of neurons that are positive at initialization. We will show that, before any $r_i$ can change sign, each neurons for which $a_j(0)y_i < 0$ reaches 0. Let $t_n^*$ be the first time any $|r_i - y_i| > \frac{y_i}{2}$. For $t \leq t_n^*$, neurons evolve monotonously depending on the sign of $a_j(0)y_i$: for $j, i$ such that $a_j(0)y_i < 0$ the neuron decreases, and for $a_j(0)y_i > 0$ the neuron increases. If $a_j(0)y_i < 0$, we have

$$
\begin{aligned}
\frac{d}{dt}\langle w_j|x_i\rangle_+ &\leq -\frac{\sqrt{|a_j(0)|^2 - ||w_j(0)||^2}|y_i|}{2n}||x_i||^2 1_{j,i} \\
\langle w_j|x_i\rangle_+ &\leq \langle w_j(0)|x_i\rangle_+ - 1_{j,i}\frac{|y_i|}{2n}||x_i||^2 \Delta t
\end{aligned}
\tag{49}
$$

where $|a_j(0)|^2 - ||w_j(0)||^2 \geq \Delta > 0$. Other wise the same equation gives for $a_j(0)y_i > 0$

$$
\begin{aligned}
\frac{d}{dt}\langle w_j|x_i\rangle_+ &\geq \frac{\sqrt{|a_j(0)|^2 - ||w_j(0)||^2}|y_i|}{2n}||x_i||^2 1_{j,i} \\
\langle w_j|x_i\rangle_+ &\geq \langle w_j(0)|x_i\rangle_+ + 1_{j,i}\frac{|y_i|}{2n}||x_i||^2 \Delta t
\end{aligned}
\tag{50}
$$

Let $\tilde{t}_n = 2n\frac{\max_{j,i}\langle w_j(0)|\bar{x}_i\rangle_+}{C_y^- C_x^- \Delta}$, if $\tilde{t}_n \leq t_n^*$, then we have extinction in finite time, i.e., the incorrectly initialized neurons have reached 0. In the meantime, neurons for which $a_j(0)y_i > 0$ will stay positive. Let us show that at $\tilde{t}_n$, residuals have almost not moved. We thus suppose $t < t_n^*$ First, we bound the second-layer neurons $a_j$ using equation (14).

$$
\begin{aligned}
\frac{d}{dt}a_j &= \frac{1}{n}\sum_{i=1}^n r_i\langle w_j|x_i\rangle_+ \\
&\leq \frac{3C_y^+}{2n}\sum_{i=1}^n \langle w_j|x_i\rangle_+
\end{aligned}
\tag{51}
$$

From equation (48), we also have the following bound.

$$
\frac{d}{dt}\langle w_j|x_i\rangle_+ \leq \frac{|a_j||r_i 1_{j,i}|||x_i||^2}{n} \leq \frac{3}{2n}C_y^+(C_x^+)^2|a_j|
\tag{52}
$$

Combining both bound give a differential equation on $a_j$ which we can solve.

$$\frac{d}{dt}|a_j| \leq \frac{K}{n}|a_j|$$

$$|a_j| \leq |a_j(0)|e^{K\frac{t}{n}} \tag{53}$$

Where $K = \frac{9}{4}(C_y^+)^2(C_x^+)^2$. We get a similar bound on the neurons.

$$\begin{aligned}
\frac{d}{dt}\max_i \langle w_j|x_i\rangle_+ &\leq \max_i \frac{d}{dt}\langle w_j|x_i\rangle_+ \\
&\leq \frac{|a_j|}{n}\max_i |r_i|\|x_i\|^2 \\
&\leq \frac{K}{n^2}\sum_{i=1}^n \langle w_j|x_i\rangle_+ \\
&\leq \frac{K}{n}\max_i\langle w_j|x_i\rangle_+ \\
\max_i\langle w_j|x_i\rangle_+ &\leq \max_i\langle w_j(0)|x_i\rangle_+ e^{K\frac{t}{n}}
\end{aligned} \tag{54}$$

The previous bounds show us that the growth of both layers' neurons are only constant for times of order $n$. Formally we have the following bound.

$$\begin{aligned}
|r_i(t) - y_i| &= \left|\frac{1}{p}\sum_{j=1}^p a_j\langle w_j|x_i\rangle_+\right| \\
&\leq \frac{1}{p}\sum_{j=1}^p |a_j|\langle w_j|x_i\rangle_+ \\
&\leq \frac{1}{p}\sum_{j=1}^p |a_j(0)|\max_i\langle w_j(0)|x_i\rangle_+ e^{2K\frac{t}{n}} \\
&\leq \frac{C_x^+}{\sqrt{d}}\max_j|a_j(0)|\max_{i,j}\left\langle\sqrt{d}w_j(0)|\bar{x}_i\right\rangle_+ e^{2K\frac{t}{n}}
\end{aligned} \tag{55}$$

Now let us note that, the norm of $w$ was taken to be independent of $d, n, p$. So, we have by rotational invariance that for any orthonormal basis of the space $(e_i)_{i=1:d}$, we have the following equality for any $\lambda > 0$.

$$\mathbb{E}\left[e^{\frac{\langle\sqrt{d}w|e_1\rangle^2}{\lambda^2}}\right] = \Pi_{i=1}^d\mathbb{E}\left[e^{\frac{\langle w|e_i\rangle^2}{\lambda^2}}\right] = \mathbb{E}\left[e^{\frac{\|w\|^2}{\lambda^2}}\right] \tag{56}$$

Thus $\|\langle\sqrt{d}w|e_1\rangle_+\|_{\psi_2} \leq \|(\|w\|_2)^2\|_{\psi_2}$, with the upper bound that doesn't depend on $d, n, p$. We now use Proposition 2.7.6 from Vershynin [2018] to conclude that

$$\mathbb{P}\left(\max_{1\leq j\leq p}|a_j(0)| \leq t\right) \geq 1 - 2e^{-\frac{ct^2}{C\|a\|_{\psi_2}^2\log(p)}}$$

$$\mathbb{P}\left(\max_{i,j}\left\langle\sqrt{d}w_j(0)|\bar{x}_i\right\rangle_+ \leq t\right) \geq 1 - 2e^{-\frac{ct^2}{C\|\langle\sqrt{d}w|\bar{x}\rangle_+\|_{\psi_2}^2\log(np)}} \tag{57}$$

where $C, c > 0$ are constants. Thus we conclude that with probability $1 - \varepsilon$, we have

$$\max_{1\leq j\leq p}|a_j(0)| \leq \|a\|_{\psi_2}\sqrt{\frac{C}{c}\log(p)\log\left(\frac{4}{\varepsilon}\right)}$$

$$\max_{i,j}\left\langle\sqrt{d}w_j(0)|\bar{x}_i\right\rangle_+ \leq \left\|\left\langle\sqrt{d}w|\bar{x}\right\rangle_+\right\|_{\psi_2}\sqrt{\frac{C}{c}\log(np)\log\left(\frac{4}{\varepsilon}\right)} \tag{58}$$

This give the new upper bound of

$$|r_i(t) - y_i| \leq \frac{C_x^+}{\sqrt{d}}\left\|(\|w\|_2)^2\right\|_{\psi_2}\|a\|_{\psi_2}\frac{C}{c}\log\left(\frac{4}{\varepsilon}\right)\sqrt{\log(np)\log(p)}e^{2K\frac{t}{n}} \tag{59}$$

Now, we pose $\bar{t}_n$ with the following definition.

$$\bar{t}_n = \frac{n}{2K} \left[ \log \left( \frac{\sqrt{d}}{\sqrt{\log(p)\log(np)\log\left(\frac{4}{\varepsilon}\right)}} \right) - \log \left( 3||a||_{\psi_2} \left||(||w||_2)^2\right||_{\psi_2} \frac{C_x^+}{C_y^-} \frac{C}{c} \right) \right] \tag{60}$$

We have that $|r_i(\bar{t}_n) - y_i| \leq \frac{C_y^-}{3} \leq \frac{y_i}{2}$. This means that $\bar{t}_n \leq t_n^*$. Finally, since we have $\tilde{t}_n \leq \frac{2n}{C_y^- C_x^- \Delta} \left||(||w||_2)^2\right||_{\psi_2} \sqrt{\frac{C}{c} \frac{\log(np)\log\left(\frac{4}{\varepsilon}\right)}{d}}$. Thus, there exists a constant $\kappa$ depending only on $\Delta$, $C_{x,y}^{+,-}$, $\left||(||w||_2)^2\right||_{\psi_2}$ and $||a||_{\psi_2}$, such that for $d \geq \kappa \log(p)\log(np)\log\left(\frac{1}{\varepsilon}\right)^2$, we have $\tilde{t}_n \leq \bar{t}_n$.

This concludes the proof, showing that the neuron initialized with $a_j y_i < 0$ reach 0 before $\tilde{t}_n$ of order $n$. $\qquad\square$

*Proof.* **of Proposition 1**
Thanks to Lemma 8, there exists $t_n$ such that for $t \geq t_n$, each example has only correctly activated neurons. Without loss of generality suppose that all labels are positive. Then the network only has positive contributions $a_j(t)\langle w_j|x_i\rangle \geq 0$ for all i. Let $N(j,i)$ the number of indices $q$ such that $a_j(t)\langle w_j|x_i\rangle \leq a_q(t)\langle w_q|x_i\rangle$. We have

$$\frac{N(j,i)}{p} a_j\langle w_j|x_i\rangle \leq \frac{1}{p}\sum_{q=1}^{p} a_q\langle w_q|x_i\rangle = y_i - r_i \tag{61}$$

Thus, we can bound the norm of $w_j$,

$$
\begin{aligned}
||w_j||^4 &\leq a_j^2||w_j||^2 \\
&= \sum_{i=1}^{n} a_j^2\langle w_j|\bar{x}_i\rangle^2 \\
&\leq \sum_{i=1}^{n} \frac{p^2}{N(j,i)^2} \left( \frac{y_i - r_i}{||x_i||} \right)^2 \\
&\leq \max_i \left( \frac{y_i - r_i}{C_x^-} \right)^2 \sum_{i=1}^{n} \frac{p^2}{N(j,i)^2}
\end{aligned}
\tag{62}
$$

This helps us upper bound the sum of $|a_j|^2 + ||w_j||^2$.

$$
\begin{aligned}
\frac{1}{p}\sum_{j=1}^{p} \left( |a_j|^2 + ||w_j||^2 \right) 1_{j,i} &\leq \frac{1}{p}\sum_{j=1}^{p} \left( |a_j(0)|^2 - ||w_j(0)||^2 \right) 1_{j,i} + \frac{2}{p}\sum_{j=1}^{p} ||w_j||^2 \\
&\leq \bar{C} + \frac{2}{p}\sum_{j=1}^{p} \frac{\max_i(y_i - r_i)}{C_x^-} \sqrt{\sum_{i=1}^{n} \frac{p^2}{N(j,i)^2}} \\
&\leq \bar{C} + 2\frac{\max_i(y_i - r_i)}{C_x^-} \sum_{j=1}^{p} \sqrt{\sum_{i=1}^{n} \frac{1}{N(j,i)^2}} \\
&\leq \bar{C} + 2\frac{\max_i(y_i - r_i)}{C_x^-} \sqrt{p\sum_{j=1}^{p}\sum_{i=1}^{n} \frac{1}{N(j,i)^2}} \\
&\leq \bar{C} + \pi\sqrt{\frac{2}{3}} \frac{\max_i(y_i - r_i)}{C_x^-} \sqrt{np}
\end{aligned}
\tag{63}
$$

Using the upper bound of equation (23) with the previous inequality gives us the upper control on $\mu(t)$.

$$\mu(t) \leq 2\pi\sqrt{\frac{2}{3}\frac{(C_x^+)^2}{C_x^-}} C_y^+ \max_i \left|1 - \frac{r_i}{y_i}\right| \sqrt{\frac{p}{n}} + \frac{2}{n}(C_x^+)^2\bar{C} \tag{64}$$

Thus, the constant $C$ from the Proposition statement is

$$C = \max \left( \frac{2}{C_y^- C_x^- \Delta} \max_{j,i} \langle w_j(0)|x_i \rangle_+, \right.$$

$$2(C_x^+)^2 \frac{1}{p} \sum_{j=1}^p \left( |a_j(0)|^2 - ||w_j(0)||^2 \right) 1_{j,i},$$

$$2\pi \sqrt{\frac{2}{3}} \frac{(C_x^+)^2}{C_x^-} C_y^+,$$

$$\left. \kappa \right)$$

(65)

with $\kappa$ the constant of Lemma 8. $\qquad\square$

### A.4 Proposition 2

*Proof.* **of Proposition** 2

Recall that we initialize the network with $p_n$ neurons, for which there are exactly $k_n$ examples positively correlated with it, *i.e.*, for $q \neq j$ that $\langle w_j|x_i^q \rangle \leq 0$ at all time. This means that we can write $h_\theta(x_i^j) = \frac{a_j}{p_n} \langle w_j|x_i^j \rangle_+ = \frac{s_j}{p_n} ||w_j||_+ \langle w_j|x_i^j \rangle_+$, and $s_j$ does not change by Lemma 1. This implies that the dynamics is decoupled: $w_j$ and $w_q$ can be studied separately.

Let us compute the dynamics for the neuron $j$. We let $D_j^n = \frac{1}{\sqrt{k_n}} \sum_{i=1}^{k_n} y_i^j x_i^j$, $R_j = \sum_{i=1}^{k_n} r_i^j x_i^j$, $||w_j||_+^2 = \sum_{i=1}^n \langle w_j|x_i \rangle_+^2$, and $\bar{x}^+ = \frac{x}{||x||_+}$. We first consider the alignment between $w_j$ and $D_j^n$:

$$\frac{d}{dt} \langle \bar{D}_j^n|s_j \bar{w}_j^+ \rangle = \left\langle \bar{D}_j^n \big| I_d - \bar{w}_j^+ (\bar{w}_j^+)^T \big| \frac{1}{||w_j||_+} \frac{d}{dt} w_j \right\rangle$$

$$= \frac{1}{n} \langle \bar{D}_j^n | I_d - \bar{w}_j^+ (\bar{w}_j^+)^T | R_j \rangle$$

$$= \frac{1}{n} \langle \bar{D}_j^n | I_d - \bar{w}_j^+ (\bar{w}_j^+)^T | k_n D_j^n - \frac{s_j}{p_n} ||w_j||_+ w_j \rangle \qquad (66)$$

$$= \frac{\sqrt{k_n} ||D_j^n||}{n} (1 - \langle \bar{D}_j^n|s_j \bar{w}_j^+ \rangle^2)$$

$$= \frac{c_n^j}{2} (1 - \langle \bar{D}_j^n|s_j \bar{w}_j^+ \rangle^2)$$

This equation has a closed-form solution which is

$$\langle \bar{D}_j^n|s_j \bar{w}_j^+ \rangle = \frac{\sinh\left(c_n^j t\right) + \langle \bar{D}_j^n|s_j \bar{w}_j^+(0) \rangle \cosh\left(c_n^j t\right)}{\cosh\left(c_n^j t\right) + \langle \bar{D}_j^n|s_j \bar{w}_j^+(0) \rangle \sinh\left(c_n^j t\right)}$$

$$= \frac{1}{c_n^j} \frac{d}{dt} \left[ \log\left( \cosh\left(c_n^j t\right) + \langle \bar{D}_j^n|s_j \bar{w}_j^+(0) \rangle \sinh\left(c_n^j t\right) \right) \right]$$

(67)

Now we can compute the norm of the neuron.

$$\frac{d}{dt} ||w_j||_+^2 = 2||w_j||_+ \frac{s_j}{n} \sum_{i=1}^{k_n} r_i \langle w_j|x_i^j \rangle_+$$

$$= \frac{2}{n} ||w_j||_+^2 (\langle D_j^n|s_j \bar{w}_j^+ \rangle - \frac{1}{p_n} ||w_j||_+^2)$$

$$||w_j||_+^2 e^{\frac{2}{np_n} \int_0^t ||w_j(u)||_+^2 du} = ||w_j(0)||_+^2 e^{\frac{2}{n} \int_0^t \langle D_j^n|s_j \bar{w}_j^+(u) \rangle du} \qquad (68)$$

$$e^{\frac{2}{np_n} \int_0^t ||w_j(u)||_+^2 du} - 1 = \frac{2}{np_n} ||w_j(0)||_+^2 \int_0^t e^{\frac{2}{n} \int_0^u \langle D_j^n|s_j \bar{w}_j^+(v) \rangle dv} du$$

$$||w_j(t)||_+^2 = \frac{||w_j(0)||_+^2 e^{\frac{2}{n} \int_0^t \langle D_j^n|s_j \bar{w}_j^+(u) \rangle du}}{1 + \frac{2}{np_n} ||w_j(0)||_+^2 \int_0^t e^{\frac{2}{n} \int_0^u \langle D_j^n|s_j \bar{w}_j^+(v) \rangle dv} du}$$

Finally, we can replace the expression of the correlation.

$$||w_j(t)||_+^2 = \frac{p_n\sqrt{k_n}||D_j^n|| \times ||w_j(0)||_+^2 \left(\cosh\left(c_n^j t\right) + \langle\bar{D}_j^n|s_j\bar{w}_j^+(0)\rangle \sinh\left(c_n^j t\right)\right)}{p_n\sqrt{k_n}||D_j^n|| + ||w_j(0)||_+^2 \left(\sinh\left(c_n^j t\right) + \langle\bar{D}_j^n|s_j\bar{w}_j^+(0)\rangle(\cosh\left(c_n^j t\right) - 1)\right)} \quad (69)$$

We use this equation in Lemma 7, and easily obtain the upper bound thanks to the monotonicity of $||w_j(t)||^2 \leq ||w_j(t=+\infty)||^2 = p_n\sqrt{k_n}||D_j^n||$.

$$\mu(t) \leq \frac{4}{np_n}\max_i \sum_j^p ||w_j(t)||_+^2 1_{j,i} = \frac{4\max_j ||w_j(t)||_+^2}{np_n} \leq \frac{4C_y^+\sqrt{k_n}}{n} = \frac{4C_y^+}{n^\alpha} \quad (70)$$

For the lower bound, we have the bound for $t \geq \frac{\alpha}{2C_y}n^{3\alpha-1}\log\left(n(C_y^+)^{\frac{1}{\alpha}}\right) \geq \frac{1}{c_n^j}\log(p_n\sqrt{k_n}||D_j^n||)$ by monotonicity. Indeed,

$$\cosh\left(c_n^j t\right) + \langle\bar{D}_j^n|s_j\bar{w}_j^+(0)\rangle \sinh\left(c_n^j t\right) \geq \frac{1}{2}e^{c_n^j t}\left(1 + \langle\bar{D}_j^n|s_j\bar{w}_j^+(0)\rangle\right) \quad (71)$$

and

$$\sinh\left(c_n^j t\right) + \langle\bar{D}_j^n|s_j\bar{w}_j^+(0)\rangle(\cosh\left(c_n^j t\right) - 1) \leq \frac{1}{2}e^{c_n^j t}\left(1 + \langle\bar{D}_j^n|s_j\bar{w}_j^+(0)\rangle\right) \quad (72)$$

which implies that

$$\begin{aligned}
||w_j(t)||_+^2 &\geq \frac{p_n\sqrt{k_n}||D_j^n|| \times ||w_j(0)||_+^2 e^{c_n^j t}\left(1 + \langle\bar{D}_j^n|s_j\bar{w}_j^+(0)\rangle\right)}{2p_n\sqrt{k_n}||D_j^n|| + ||w_j(0)||_+^2 e^{c_n^j t}\left(1 + \langle\bar{D}_j^n|s_j\bar{w}_j^+(0)\rangle\right)} \\
&\geq p_n\sqrt{k_n}||D_j^n||\frac{||w_j(0)||_+^2\left(1 + \langle\bar{D}_j^n|s_j\bar{w}_j^+(0)\rangle\right)}{2 + ||w_j(0)||_+^2\left(1 + \langle\bar{D}_j^n|s_j\bar{w}_j^+(0)\rangle\right)} \\
&\geq p_n\sqrt{k_n}||D_j^n||\frac{||w_j(0)||_+^2}{2 + ||w_j(0)||_+^2}
\end{aligned} \quad (73)$$

since $\langle\bar{D}_j^n|s_j\bar{w}_j^+(0)\rangle \geq 0$. Finally, we obtain the desired lower bound.

$$\mu(t) \geq \frac{2}{np_n}\min_i \sum_j^p ||w_j(t)||_+^2 1_{j,i} = \frac{2\min_j ||w_j(t)||_+^2}{np_n} \geq \frac{2C_y^-}{n^\alpha}\min_j \frac{||w_j(0)||_+^2}{2 + ||w_j(0)||_+^2} \quad (74)$$

$\square$

## A.5 Lemma 4

*Proof.* **of Lemma 4**
Recall that, in this proof, $p$ and $n$ are independent parameters of the system. Let us recall the equation of the local-PL curvature on the system that was found in equation (16).

$$\begin{aligned}
\mu(0) &= \frac{2}{np}\sum_{j=1}^p (w_j(0)^T X P_j R(0))^2 + |a_j(0)|^2 ||X P_j R(0)||^2 \\
&= \frac{2}{n}R(0)^T\left(\frac{1}{p}\sum_{j=1}^p P_j^T X^T w_j(0)w_j^T(0)X P_j + |a_j(0)|^2 P_j^T X^T X P_j\right)R(0) \\
&= \frac{2}{n}R(0)^T M(0)R(0)
\end{aligned} \quad (75)$$

Recall that we note $P_j$ the diagonal matrix with diagonal $1_{j,i} = \mathbb{1}_{\langle w_j|x_i\rangle_+ > 0}$. This means that since all terms in the sums can be computed from $a_j$ and $\langle w_j|x_i\rangle_+$, which are mutually independent random variable, that variables in the sum are mutually independent as well, and by Central Limit Theorem we have

$$M(0) = \mathbb{E}_{w,a}\left[P^T X^T w w^T X P + |a|^2 P^T X^T X P\right] + \frac{\zeta_p}{\sqrt{p}},$$

where the expectancy is taken over the neurons of the network, but not over the data $X$ and $Y$, and

$$\zeta_p \xrightarrow[p \to +\infty]{} \zeta \sim \mathcal{N}\left(0, \mathbb{V}_{w,a}(P^T X^T w w^T X P + |a|^2 P^T X^T X P)\right) = \mathcal{N}(0, \mathbb{V}(\zeta)).$$

We now apply the Central Limit Theorem on the residual $R(0)$.

$$R(0) = Y - \frac{1}{p}\sum_{j=1}^{p} a_j P_j X^T w_j = Y - \mathbb{E}_{w,a}[a P X^T w] - \frac{\xi_p}{\sqrt{p}} = \tilde{Y} - \frac{\xi_p}{\sqrt{p}} \tag{76}$$

with $\xi_p \xrightarrow[p \to +\infty]{} \xi \sim \mathcal{N}\left(0, \mathbb{V}_{w,a}(a P X^T w)\right) = \mathcal{N}(0, \mathbb{V}_{w,a}(\xi))$. Now, using equations 75 and 76, we have that

$$\mu(0) \xrightarrow[p \to +\infty]{} \frac{2}{n}\mathbb{E}_{w,a}\left[(w^T X P \tilde{Y})^2 + |a|^2 \|X P \tilde{Y}\|^2\right] = \frac{\beta_0}{n} \tag{77}$$

and for the next order

$$\sqrt{p}\,(n\mu(0) - \beta_0) \xrightarrow[p \to +\infty]{} \mathcal{N}(0, \gamma_0^2) \tag{78}$$

with

$$\gamma_0^2 = \tilde{Y}^T \mathbb{V}(\zeta)\tilde{Y} + 2\mathbb{V}(\xi)\mathbb{E}_{w,a}\left[P^T X^T w w^T X P + |a|^2 P^T X^T X P\right]\tilde{Y} \tag{79}$$

$\square$

## A.6  Theorem 2

*Proof.* **of Theorem 2**
We consider the setting of Proposition 2 but with a fixed number of neuron $p$, and as in its proof, we focus on one specific neuron $j$ for which we suppose $s_j = 1$. We can rewrite equation 69.

$$\|w_j(t)\|_+^2 = \frac{p\sqrt{k_n}\|D_j^n\|\|w_j(0)\|_+^2 \left(\cosh\left(c_n^j t\right) + \langle \bar{D}_j^n|\bar{w}_j^+(0)\rangle \sinh\left(c_n^j t\right)\right)}{p\sqrt{k_n}\|D_j^n\| + \|w_j(0)\|_+^2 \left(\sinh\left(c_n^j t\right) + \langle \bar{D}_j^n|\bar{w}_j^+(0)\rangle(\cosh\left(c_n^j t\right) - 1)\right)} \tag{80}$$

Let us rewrite the loss of the group $j$.

$$\begin{aligned} L^j(t) &= \frac{1}{2k_n}\sum_{i=1}^{k_n}(r_i^j)^2 \\ &= \frac{1}{2k_n}\sum_{i=1}^{k_n}\left(y_i^j - \frac{\|w_j\|_+}{p}\langle \bar{w}_j^+|x_i^j\rangle\right)^2 \\ &= \frac{1}{2}\left[\|D_j^n\|^2 - \frac{2}{\sqrt{k_n}p}\|D_j^n\|\langle \bar{D}_j^n|\bar{w}_j^+\rangle\|w_j\|_+^2 + \frac{1}{k_n p^2}\|w_j\|_+^4\right] \end{aligned} \tag{81}$$

Let $t_n^j(\kappa) = \frac{1}{c_n^j}\log(\kappa p\sqrt{k_n}\|D_j^n\|)$, where $c_n^j = \frac{2\sqrt{k_n}\|D_j^n\|}{n}$, which depends on the variable $\kappa > 0$. We have

$$\|w_j(t_n^j(\kappa))\|_+^2 = p\sqrt{k_n}\|D_j^n\| \frac{\kappa\|w_j(0)\|_+^2 \left(1 + \frac{K(j,n)^2}{\kappa^2} + \langle \bar{D}_j^n|\bar{w}_j^+(0)\rangle\left(1 - \frac{K(j,n)^2}{\kappa^2}\right)\right)}{2 + \kappa\|w_j(0)\|_+^2 \left(1 - \frac{K(j,n)^2}{\kappa^2} + \langle \bar{D}_j^n|\bar{w}_j^+(0)\rangle\left(1 - \frac{K(j,n)}{\kappa}\right)^2\right)} \tag{82}$$

with $K(j,n) = \frac{1}{\sqrt{k_n}p\|D_j^n\|}$. Moreover, we have

$$\langle \bar{D}_j^n|s_j\bar{w}_j^+(t_n^j(\kappa))\rangle = \frac{1 - \frac{K(j,n)^2}{\kappa^2} + \langle \bar{D}_j^n|\bar{w}_j^+(0)\rangle\left(1 + \frac{K(j,n)^2}{\kappa^2}\right)}{1 + \frac{K(j,n)^2}{\kappa^2} + \langle \bar{D}_j^n|\bar{w}_j^+(0)\rangle\left(1 - \frac{K(j,n)^2}{\kappa^2}\right)}. \tag{83}$$

Thus, by taking $n$ large enough, there exists $\kappa(j,n,\varepsilon)$ such that $L^j(t_n^j(\kappa(j,n,\varepsilon))) = L^j(t_n^j(\varepsilon)) = \varepsilon\|D_j^n\|^2$. For simplification, we use $t_n^j(\varepsilon) = t_n^j(\kappa(j,n,\varepsilon))$. Moreover, $\kappa(j,n,\varepsilon) \to \kappa^j(\varepsilon)$ when $n$ goes to infinity.

$$L^j(t_n^j(\varepsilon)) \to \frac{1}{2}\left[\|D_j^\infty\| - \frac{\kappa^j(\varepsilon)\|w_j(0)\|_+^2}{2 + \kappa^j(\varepsilon)\|w_j(0)\|_+^2}\right]^2 = \frac{\varepsilon}{2}\|D_j^\infty\|^2 \tag{84}$$

This shows that $\kappa^j(\varepsilon) = \frac{2}{||w_j(0)||_+^2} \frac{||D_j^\infty||(1-\sqrt{\varepsilon})}{1+||D_j^\infty||(1-\sqrt{\varepsilon})}$. Thus, we have a phase transition since the lost goes from $1-\varepsilon$ to $\varepsilon$ in a time which, after normalization, goes to 0. The time of the phase transition is

$$\frac{t_n^j(\varepsilon)}{t_n} = 4\frac{\log(\kappa(j,n,\varepsilon)p\sqrt{k_n}||D_j^n||)}{c_n^j\sqrt{np}\log(n)} \sim_n \frac{1}{||D_j^\infty||} \tag{85}$$

and its cutoff window is

$$\frac{t_n^j(\varepsilon) - t_n^j(1-\varepsilon)}{t_n} = \log\left(\frac{\kappa(j,n,\varepsilon)}{\kappa(j,n,1-\varepsilon)}\right)\frac{1}{c_n^j t_n} \sim_n \frac{1}{2||D_j^\infty||}\log\left(\frac{\kappa^j(\varepsilon)}{\kappa^j(1-\varepsilon)}\right)\frac{1}{\log(n)} \tag{86}$$

where we recall that $t_n = \frac{\sqrt{np}}{4}\log(n)$. We conclude that the normalized loss thus has at most $p$ phase transition at times $\frac{1}{||D_j^\infty||}$. Moreover, the constant in the Theorem is

$$C^j(\varepsilon) = \frac{(1-\sqrt{\varepsilon})}{1+||D_j^\infty||(1-\sqrt{\varepsilon})}\frac{1+||D_j^\infty||\left(1-\sqrt{1-\varepsilon}\right)}{\left(1-\sqrt{1-\varepsilon}\right)} \sim \frac{1}{\varepsilon}\frac{2}{1+||D_j^\infty||} \tag{87}$$

$\square$

### A.7 Other results

*Proof.* **of Lemma 1**
We verify that $\langle \frac{d}{dt}w_j | w_j \rangle = a_j \frac{d}{dt}a_j$, using the derivations from equations (14) and (13). Integrating this equality gives the expected invariance. $\square$

*Proof.* **of Lemma 2**
By definition, we have $\mu(t) = \frac{|\nabla L(\theta_t)|^2}{L(\theta_t)}$, and by property of the gradient flow, $|\nabla L(\theta_t)|^2 = -\frac{d}{dt}L(\theta_t)$. Thus,

$$\frac{d}{dt}L(\theta_t) = -\mu(t)L(\theta_t)$$
$$\log(L(\theta_t)) - \log(L(\theta(0))) = -\int_0^t \mu(x)dx \tag{88}$$
$$L(\theta_t) = L(\theta(0))e^{-\langle\mu(t)\rangle t}$$

$\square$

## B  Experiments

This Appendix contains additional details on the experiments done in Section 5. Data generation and weight initialization were performed as follows: we initialize all neurons independently as $w_j \sim \mathcal{N}(0, \frac{1}{d}I_d)$ as well as $\frac{a_j}{|a_j|} \sim \mathcal{U}(\{-1,1\})$ and $|a_j| - ||w_j|| \sim \text{Exp}(1)$ which implies $|a_j(0)| \geq ||w_j(0)||$. For the data, we consider $y_i \sim \mathcal{U}([-2,-1] \cup [1,2])$ and $||x_i|| \sim \mathcal{U}([1,2])$ in order to control the constants $C_x^-, C_y^- \geq 1$ and $C_x^+, C_y^+ \leq 2$. Finally, in order to fall within the assumptions of Lemma 3, we let $\frac{x_i}{||x_i||} \sim \mathcal{U}(\mathbb{S}^{d-1})$ in Section 5.1, and $\frac{x_i}{||x_i||}$ be an orthogonal family in Section 5.2.

**Experiment 1.** For the experiment in Figure 3, we trained 500 networks in dimension 100, with $n$ between 2500 and 3500, with 25 runs for each value of $n$. We used $p = \lfloor\frac{\log(\frac{n}{\varepsilon})}{\log(\frac{4}{3})}\rfloor + 1$ neurons for each experiment with $\varepsilon = 0.05$, since this is the optimal threshold obtained in Lemma 5. We trained the networks with gradient descent using a learning rate of 1 for a total time $t_\infty = 1.5 \times \frac{\sqrt{np}}{4}\log(np)$ and thus $e = \frac{t_\infty}{\text{lr}}$ epochs.

We considered that a network converged as long as its loss went below $\frac{C_y^-}{2n}$, which then guarantees convergence to 0. We thus early stopped the training and declared the loss was exactly 0. Otherwise, the convergence went for all epochs and the network was assumed to not be able to reach 0 loss. In

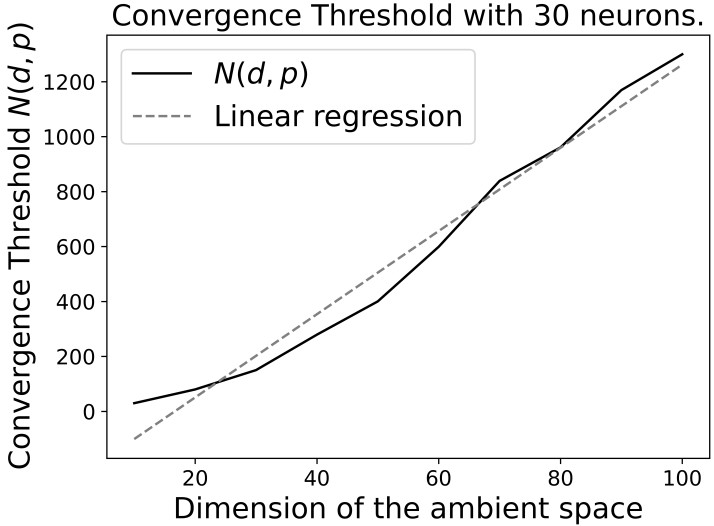

Figure 4: This graph shows the scaling law of the convergence threshold for a fixed number of neurons. It suggests that the scaling is linear in $d$: $N(d, p) = C(p)d$.

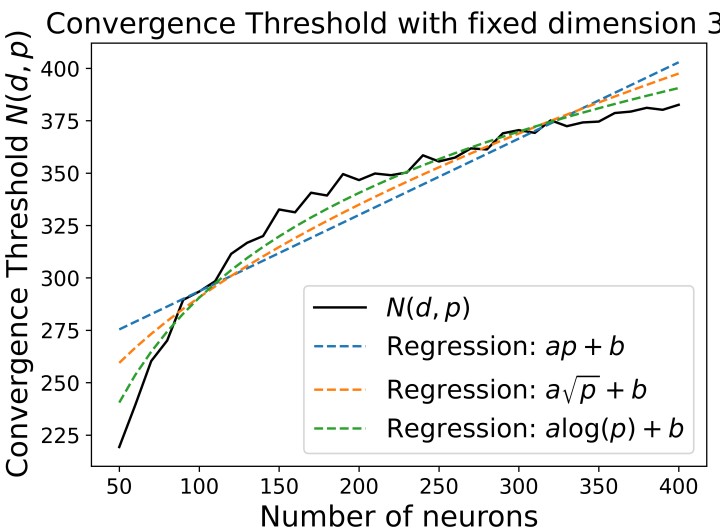

Figure 5: This graph show the scaling law of the convergence threshold for a fixed number of neurons. It suggests that the scaling is not linear in $d$, but it is hard to differentiate between a sub-linear polynomial growth or a logarithmic growth.

doted line, we interpolate the probability plot using a sigmoid function, and learned automatically the convergence threshold $N(d, p)$.

For the scaling law on $d$, we fixed $p$ at 30, and trained networks with dimension varying from 10 to 100, and $n$ ranging from $N(d, p) - 15d$ to $N(d, p) + 15d$, with step $d$. For each dimension, we interpolate the probability graph using a sigmoid, and plotted the linear trend on Figure 4. For the scaling in $p$, we fixed $d = 30$, and varied $p$ from 50 to 400, and plotted the trend on Figure 5 which shows that the scaling in $p$ is sub-linear.

**Experiment 2.** For this experiment, we trained 500 networks in dimension 2000, with $n$ between 1000 and 2000, with 25 runs for each value of $n$. We used the same number of neurons, learning rates, and epochs as in experiment 1. Let us recall the 4 measures we plotted on the Figure 6:

Figure 6: Scaling laws in log-log for different measures of the local-PL curvature in dimension 2000. Each curve is in fact linear with slope close to $-\frac{1}{2}$, which is expected by Conjecture 1.

1. The instantaneous local-PL curvature at the end of the training, $\mu(t_\infty) = \log\left(\frac{L(t_\infty - 1)}{L(t_\infty)}\right)$,

2. The average-PL curvature throughout the training, $\langle\mu_\infty\rangle = \log\left(\frac{L(0)}{L(t_\infty)}\right)$,

3. The lower bound on the local-PL at the end of the training, $\mu_{\text{low}} = \frac{2}{n}\min_i \frac{1}{p}\sum_{j=1}^{p}|a_j|^2 1_{j,i}$,

4. The upper bound on the local-PL at the end of the training, $\mu_{\text{upp}} = \frac{16}{n}\max_i \frac{1}{p}\sum_{j=1}^{p}|a_j|^2 1_{j,i}$.

Each of the slope being close to $-\frac{1}{2}$, we conclude from this log-log graph that $\langle\mu_\infty\rangle = \frac{K}{\sqrt{n}}$ as foreseen in Conjecture 1.

Each plot's related experiments were performed on a MacBook Air under 2 hours without acceleration materials.

## C   Additional results

### C.1   Collapse of the second layer

Similar to the early alignment phenomenon described by Boursier and Flammarion [2024a,b, Theorem 2], where the neurons can rotate and collapse to align on a single vector preventing minimization of the loss, the weights $a_j$ of the second layer can also collapse on a single direction. Under the hypothesis that $|a_j(0)| \geq ||w_j(0)||$, the scalar $a_j$ cannot change sign, which prevents this scenario in the article's results. But if $|a_j(0)| < ||w_j(0)||$, they can change sign, and prevent global minimization even when the neurons are correctly initialized. Proposition 3 gives an example of such collapse in low dimension.

**Proposition 3.** *Suppose that $d = n = p = 2$. Let $(x_1, x_2)$ be the canonical basis of $\mathbb{R}^2$, with the outputs satisfying $y_1 y_2 < 0$, $\lambda = |\frac{y_2}{y_1}|$. Let $|a_1(0)|, |a_2(0)| \leq \delta$, and let $\min_{j,i}\langle w_j(0)|x_i\rangle > 0$. Then, for $\delta$ small enough, $y_1$ large enough, and*

$$\sqrt{\max_{j,i}\langle w_j(0)|x_i\rangle \frac{8}{y_1}} \leq \lambda \leq \frac{\min_{j,i}\langle w_j(0)|x_i\rangle}{\max_{j,i}\langle w_j(0)|x_i\rangle} \tag{89}$$

*we have* $\lim_{t \to +\infty} L(\theta_t) > 0$.

The proof relies on the ratio between outputs being large $\lambda$, in order to steer the $a_j$ to change signs, but not too large to then make the neuron go extinct before the signs of $a_j$ may change again. This traps the network in a state of sub-optimal loss, and if $a_j$ were initialized as large as the vectors, this collapse could not have happened.

*Proof.* **of Proposition 3** Without loss of generality, let us suppose $y_1 > 0$ and $y_2 < 0$. We will show that there are values of $\lambda, \varepsilon$ for which the system will not converge. The derivatives of $a_j$ at the beginning of the dynamics writes

$$\frac{d}{dt}a_j = \frac{1}{4}(r_1\langle w_j|x_1\rangle + r_2\langle w_j|x_2\rangle)$$

$$\left|\frac{d}{dt}a_1 - \frac{y_1}{4}(\langle w_1|x_1\rangle - \lambda\langle w_1|x_2\rangle)\right| \leq \max(|a_1|,|a_2|)\max(||w_1||,||w_2||) \tag{90}$$

$$\left|\frac{d}{dt}a_2 - \frac{y_1}{4}(\langle w_2|x_1\rangle - \lambda\langle w_2|x_2\rangle)\right| \leq \max(|a_1|,|a_2|)\max(||w_1||,||w_2||)$$

and the derivatives of $\langle w_j|x_i\rangle$ are

$$\frac{d}{dt}\langle w_j|x_i\rangle = \frac{a_j r_i \mathbb{1}_{j,i}}{2}$$

$$\left|\frac{d}{dt}\langle w_j|x_1\rangle - \frac{a_j y_1 \mathbb{1}_{j,1}}{2}\right| \leq \max(|a_1|,|a_2|)\max(||w_1||,||w_2||) \tag{91}$$

$$\left|\frac{d}{dt}\langle w_j|x_2\rangle + \lambda\frac{a_j y_2 \mathbb{1}_{j,2}}{2}\right| \leq \max(|a_1|,|a_2|)\max(||w_1||,||w_2||)$$

Now suppose that for $t \leq T$, $|a_j|, ||w_j|| \leq M$ and $\langle w_j|x_i\rangle \geq m > 0$, we have

$$a_j(t) \geq -\delta + \left(\frac{y_1}{4}(m - \lambda M) - M^2\right)t$$

$$a_j(t) \leq \delta + \left(\frac{y_1}{4}(M - \lambda m) + M^2\right)t \tag{92}$$

Thus, for $T > \frac{\delta}{\left(\frac{y_1}{4}(m-\lambda M)-M^2\right)}$ with $\lambda \geq \frac{m}{M}$ and $y_1 \geq \frac{4M^2}{m-\lambda M}$, we have $a_j(T) > 0$. We now wish to find the constants $M, m$ such that the previous equation will hold. To find the constraint on $m$ and $M$, let us write

$$\langle w_j|x_1\rangle \geq \langle w_j(0)|x_1\rangle - \left(\frac{1}{2}y_1 M + M^2\right)t$$

$$\langle w_j|x_2\rangle \geq \langle w_j(0)|x_2\rangle - \left(\frac{\lambda}{2}y_1 M + M^2\right)t$$

$$\langle w_j|x_1\rangle \leq \langle w_j(0)|x_1\rangle + \left(\frac{1}{2}y_1 M + M^2\right)t \tag{93}$$

$$\langle w_j|x_2\rangle \leq \langle w_j(0)|x_2\rangle + \left(\frac{\lambda}{2}y_1 M + M^2\right)t$$

Thus, the constraints are

$$m \geq \min\left(\min_{j,i}\langle w_j(0)|x_2\rangle, \delta\right) - \delta\frac{2y_1 M + 4M^2}{y_1(m - \lambda M) - 4M^2}$$

$$M \leq \max\left(\max_{j,i}\langle w_j(0)|x_2\rangle, \delta\right) + \delta\frac{2y_1 M + 4M^2}{y_1(m - \lambda M) - 4M^2} \tag{94}$$

We see that the constraint are satisfied with $m \geq \min_{j,i}\langle w_j(0)|x_2\rangle - 2\delta > 0$ and $M \leq \max_{j,i}\langle w_j(0)|x_2\rangle + 2\delta$ if: $\delta$ is small enough, $y_1$ is large enough, and $\lambda < \frac{\min_{j,i}\langle w_j(0)|x_2\rangle}{\max_{j,i}\langle w_j(0)|x_2\rangle}$. Thus, there exists $T > 0$ such that at time $T$, we have $a_1(T), a_2(T) > 0$, and no neurons went extinct.

Now, let us show that neurons $\langle w_j|x_2\rangle$ will go to 0 for some time $\mathcal{T} > 0$, while the neurons $a_j$ stay positive. We can use the same equations as before, with this time $|a_j|, \|w_j\| \leq N$ for $t \leq \mathcal{T}$, and get

$$\langle w_j|x_2\rangle \leq \langle w_j(0)|x_2\rangle - \left(\frac{\lambda}{2}y_1 N - N^2\right)t \tag{95}$$

Thus, for $\mathcal{T} = \frac{2\max_j\langle w_j(0)|x_2\rangle}{\lambda y_1 N - 2N^2}$ and $\lambda y_1 \geq 2N$, we have extinction of the neurons. To find the constraint on $N$, use the bounds on the growth of $a_j$ and $\langle w_j(0)|x_1\rangle$. The constraint is

$$N \leq \max\left(\max_j\langle w_j(0)|x_1\rangle, \delta\right) + 2\max_j\langle w_j(0)|x_2\rangle\frac{y_1 + 2N}{\lambda y_1 - 2N} \tag{96}$$

Thus, the constraints are satisfied with $N \leq \max_{j,i}\langle w_j(0)|x_i\rangle(1+\frac{3}{\lambda})$ as long as $y_1$ is large enough, and

$$\lambda \geq \sqrt{\max_{j,i}\langle w_j(0)|x_i\rangle\frac{8}{y_1}}. \tag{97}$$

After time $\mathcal{T}$, the neurons $\langle w_j(0)|x_2\rangle$ went extinct, and thus we have $L(\theta_t) \geq \frac{y_2^2}{4} > 0$. $\qquad\square$

## C.2 Non-uniqueness of the gradient flow

Since $\sigma = $ReLU is non differentiable at 0, the gradient flow equation might have multiple solution for a single initialization. To see this, let us take the example of orthogonal data, we have for every neuron $w_j$ and data $x_i$, $\frac{d}{dt}\langle w_j|x_i\rangle_+ = \frac{r_i a_j}{n}\|x_i\|^2 1_{i,j}$. As long as $\langle w_j|x_i\rangle_+ > 0$, then $1_{i,j} = 1$ and the neuron can change, and if $\langle w_j|x_i\rangle < 0$, then $1_{i,j} = 0$, so the neuron doesn't change anymore, and cannot become active. We are interested in the case when $\langle w_j|x_i\rangle = 0$, which is the non-differentiable point of ReLU.

Suppose that for $t \in [t_1, t_2]$, we have $a_j(t)r_i(t) > 0$ and $\langle w_j(t_1)|x_i\rangle = 0$. Then for each $\tilde{t} \in [t_1, t_2[$ there exist trajectories $\theta^{\tilde{t}}$ such that at $1_{\langle w_j(t)|x_i\rangle} \underset{t\to\tilde{t}^+}{\to} 1$ and $1_{\langle w_j(\tilde{t})|x_i\rangle} = 0$. This means that there exist trajectories such that the neuron $\langle w_j|x_i\rangle$ start growing from $\tilde{t}$ even if the neuron was previously deactivated. The trajectory $\theta^{\tilde{t}}$ is as follow: for $t < \tilde{t}$, $\theta^{\tilde{t}}(t) = \theta^{\tilde{t}}(t_1)$, and then $\theta^{\tilde{t}}$ solve the gradient flow equation without $1_{i,j}$ in the derivative of $\langle w_j|x_i\rangle$.

Although there are different possible trajectories for a single initialization of the neurons, only one of them is realistic in the sense that it is represent what happens in practice: the trajectory where neuron don't reactivate alone, which is the limit trajectory of the trajectories from the gradient descent for small step-size.

