# OpenReview forum: "Convergence of the Gradient Flow for Shallow ReLU Networks on Weakly Interacting Data"
_NeurIPS.cc/2025/Conference — NeurIPS 2025 poster_

### Official Review · Reviewer_usDV · 2025-06-26

**Clarity:** 3
**Significance:** 2
**Originality:** 3
**Rating:** 4
**Confidence:** 4

**Summary:**

The paper establishes convergence guarantees for one-hidden-layer ReLU Networks, trained on high- dimensional data. Based on gradient flow, a continuous-time version of gradient descent, they analyse
the local Polyak-Lojasiewicz (PL) curvature of the training dynamics. Establishing a lover bound of
the local PL-curvature, they show that for weakly-correlated inputs, at least 4 log(n/ε) hidden-layer
neurons and under some initialization constraints (the second layer weights have larger or equal norm
than the weights from the first layer), the loss converges to zero with exponential rate of at least 1/n
with probability 1−ε. Considering the orthogonal case in particular, they can establish an upper
bound, ensuring a rate of convergence between 1/n and 1/√n and show empirically that a rate of
1/√n is met with high probability. Lastly they show that in the case of group initialization, there
are abrupt phase transitions between an initial convergence rate of 1/n, intermediate rates, and the
asymptotic convergence rate 1/√n.

**Questions:**

1- I am a bit confused with the rates in Theorem 1: in my understanding, the rates slow down, when n increases, and when the norm of data grows fast (role of C);  can the authors elaborate more about these points?

2-In line 64--66, authors mentioned that ``...to understand a simplified version of the optimization dynamics of this neural network, we study the continuous-time limit of gradient descent.'' Could authors please elaborate why and how this setting and the corresponding results can be adapted to discrete time setting which happens in practice?

3-I am confused with the role of p in equation 3? Could authors please elaborate more on that?


4. Also, according to results in the literature, local minima exist in shallow ReLU networks. So, what is the main message of the paper? Is it claiming that such suboptimal points do not exist, or that the optimization process escapes them?

**Ethical Concerns:**

["NO or VERY MINOR ethics concerns only"]

**Final Justification:**

My main concerns are answered and I would keep my rating.

**Limitations:**

No limitation

**Paper Formatting Concerns:**

No formatting concern

**Quality:**

3

**Strengths And Weaknesses:**

The authors address an important topic of convergence dynamics of shallow ReLU networks in more
realistic conditions than in previous works. They explain their results in details. Especially giving a sketch of proof for the main results makes it a lot more understandable and intuitive. Further, by giving an example of a setting in which their assumptions are
satisfied, or explaining the implications of their results, they well illustrate their work and it’s relevance.

---

> ### Author Rebuttal · Authors · 2025-07-31
>
> **Response**: Thank you for your review and questions, which we address below. Please feel free to follow up if any of our responses do not fully resolve your concerns.
>
> **1. Theorem 1**. The convergence rate decreasing with $n$ is due to the orthogonality constraint. Note that $n$ could not grow larger than $d$, and hence the limit $n \to \infty$ (which would break down the result due to the infinitely slow convergence) cannot happen.  Moreover, imagine the case $p=1$, we have for all $i$ that $y = ||w||<w|x_i>$, where the $x_i$ are orthonormal. This implies $||w||^2 = \sqrt{n}y$, and thus the PL is at most of order $\frac{1}{\sqrt{n}}$.
>
> **2. Gradient Descent**. The gradient flow corresponds to the limit of a gradient descent with the step size going to 0. In the small but positive step size limit, any equation of the form $\frac{d f(t)}{dt} = c(t)$ becomes $\frac{f(t + s) - f(t)}{s} = c(t) \sim c(t + s)$ with $s$ the step size. Now, we can carry the same computations and inequalities as before to find a valid discrete equivalent to Theorem 1, up to a constant depending on $s$. Taking $s$ small enough guarantees the convergence, but $s$ might need to depend on $p, n, d$ or the constants $C_{x,y}^{+,-}$.
>
> **3. Gradient Flow Equation**. This is indeed quite confusing (even for the author up until writing the paper), and comes from the normalization of the network. We wanted to find a way to normalize both the network $h$ and the loss $L$ in order to have consistent definition if $p$, $n$ or $d$ would go to infinity. We thus introduce the $\frac{1}{p}$ before the summation in $h$, which introduces a temporal dilatation: the more neurons, the slower each neuron learns. To have learning in the regime $p\rightarrow+\infty$, we thus rescale the gradient descent equation by a factor $p$. Note that this is strictly equivalent to having the normal equation without $p$, a network without the normalization, and weights initialized with norm $\frac{1}{\sqrt{p}}$, which is the default equations used in pytorch.
>
> **4. Local Minima**. Local minima exist in this setup, but the paper claims that in average, we will avoid them all. This works in two steps: first, with enough neurons, we put ourselves into a high probability zone of the parameter space which do not trap us into local minima (example: all the $|a_j|$ are positive, hence no negative label can be fitted), and second, the near-orthogonality hypothesis says that we don’t have local minima created from the weights moving around and interacting (see the example in appendix C.1). In particular, the local-PL viewpoint means that although the local-minima exist, they are avoided on the trajectory starting from the high probability zone.

---

> > ### Comment · Reviewer_usDV · 2025-08-03
> >
> > Thank you to the authors for their response.
> > Overall, I like the paper; however, some of my questions remain open:
> >
> > **Theorem 1**: For a fixed dimension d, I am still wondering why a smaller dataset X_2 (consider two datasets X_1 and X_2, with n_1 > n_2) would yield better rates.
> > Additionally, my question regarding the role of the constant C remains unanswered.
> >
> > **Gradient Flow Equation:** I am still not fully convinced about the role of \( p \). Does it imply any change in the parameter space under consideration? I believe a complete and detailed explanation is needed in the paper to clarify this point.
> >
> > **Local Minima:** I think this is a very important aspect and requires a clear and detailed discussion. In neural networks, there are infinitely many local minima (as well as global minima) with the same objective value due to rescaling properties. I am wondering how it is possible to claim that "with enough neurons, we put ourselves into a high-probability zone of the parameter space which does not trap us in local minima"?
> >
> > Also, regarding your sample: it seems that \( n = d \) is assumed there. Isn’t this in contrast with the assumptions of the paper? I find this part confusing and would appreciate some clarification.

---

> > > ### Author Response · Authors · 2025-08-05
> > >
> > > Thank you for your response and further questions. Please find below some precisions regarding your open questions.
> > >
> > > 1. **Theorem 1.** About the dependency in $n$. Technically speaking, the fact that the more data, the slower the convergence rate is shown by the orthogonal case for which the rate $\frac{1}{n}$ is tight. Intuitively, this comes from the fact that due to the orthogonality, the system is equivalent to solving $n$ independent one neuron problem whose individual exponential rate of convergence is $\frac{K}{n}$ with $K$ that can be found in the proof of Proposition 2 (equation (55) for an upper bound of K, and (59) for a lower bound). Because we normalize the loss for independent problems, we have an artificially slower convergence rate.
> > > 2. **Role of C.** Sorry to have missed that part of your question. In Theorem 1, $C = \frac{C_x^-}{C_x^+}C_x^-C_y^-$ which means that the dynamic is faster when the norm of the data is larger. When the data has a large norm, then every gradient has a large norm too, see equation (74) which depends on $||x_i||$.
> > > 3. **Gradient flow equation.** A short answer is that this scaling in $p$ corresponds to the scaling for which the limit $p \to \infty$ is non-degenerate and leads to a mean-field system. This is explained clearly in Theorem 1 of Chizat & Bach review article for the ICM [1] (see the scaling in equation 3.4). Concretely speaking, we could also prove in Appendix that the normalization we are using is equivalent to the usual one from Pytorch. Just to make the claim clear, here is the statement: Let $\theta = (w_j, a_j) \mathbb{R}^{d+1 \times p}$ and $\tilde{\theta} = (\tilde{w_j}, \tilde{a_j}) = (\frac{w_j}{\sqrt{p}}, \frac{a_j}{\sqrt{p}})$; let $h(x) = \frac{1}{p}\tilde{h}(x) = \frac{1}{p}\sum_{j=1}^pa_j\sigma(\langle w_j|x_i\rangle)$; let $\theta_t$ a solution to $\frac{d}{dt}\theta_t = -p\nabla_{\theta_t}L(\theta_t)$ and $\tilde{\theta}_t$ a solution to $\frac{d}{dt}\tilde{\theta}_t = -\nabla _{\tilde{\theta}_t}\tilde{L} (\tilde{\theta}_t) $ where $L$ is the loss with $h$ and $\tilde{L}$ the loss with $\tilde{h}$. Then we have $\theta_t = \tilde{\theta}_t$. Now the $\sim$ gradient flow is the one from Pytorch, but hides the normalization in $p$ inside the initialization to not accelerate the dynamic by $p$ inside the gradient descent.
> > > 4. **Local minima.** We are able to avoid local minima because, in this scenario where the examples satisfy an orthogonality constraint, no local minima are found *after initialization*. That is, if the network is correctly initialized (e.g., according to Lemma 5), it will converge, since the interactions between examples are small enough to prevent new local minima from forming during training. In this setting, either the network gets stuck at the beginning and never attempts to learn a particular example $x_i$, or it successfully learns it.
> > > 5. **Samples.** In the experiments, we train networks in order to find the limit for which Corollary 1 will empirically hold, and we are moreover interested in the scaling between $n$ and $d$ rather than absolute values. This is why we take a higher number of samples $n$ and dimensions $d$ although this is against our Theorem 1 requirements: empirically, the networks converges with high probability well beyond the theoretical bound, which let us hope for future improvement in this bound.
> > >
> > > [1]: Bach, F. and Chizat, L., 2021. Gradient descent on infinitely wide neural networks: Global convergence and generalization. *arXiv preprint arXiv:2110.08084*.

---

> > > > ### Comment · Reviewer_usDV · 2025-08-06
> > > >
> > > > Thanks to the authors for providing additional details; this has clarified things better.

---

### Official Review · Reviewer_5dzT · 2025-06-30

**Clarity:** 3
**Significance:** 2
**Originality:** 3
**Rating:** 4
**Confidence:** 3

**Summary:**

This paper studies the convergence rate of two-layers ReLU networks on "almost" orthogonal data. The authors first show a convergence rate of $\Omega(1/n),$ for the weak-correlated data case, where the correlation of the data points are of order $O(1/\sqrt{n}).$ Then the authors show the upper bound of convergence rate is of order $\O(1/\sqrt{n})$ for completely orthogonal data, and $O(1/n^\alpha)$ for any $alpha \in [1/2, 1]$ under extra initialization assumptions. Finally, the authors show the different stages of training under the settings with completely orthogonal data and the initialization assumptions.

**Questions:**

1. Please also refer to the Strengths and Weaknesses part.

2. For the results in Proposition 2, what are the technical difficulties that require the completely  orthogonal data?

3. For the group initialization, is it possible to prove it for any reasonable initialization of the weights? For example, random initialization with small enough variance.

**Ethical Concerns:**

["NO or VERY MINOR ethics concerns only"]

**Final Justification:**

This paper studies the convergence rate of two-layers ReLU networks on "almost" orthogonal data. After the rebuttal. My initial major concern is on the strong group initialization, and a feww other assumptions. In the rebuttal, the authors mentioned that it is possible to generalize the results to Gaussian initialization with small enough variance, which justifies the utility of the theoretical results better in my opinion. Thus, I would like to keep my initial evaluation.

**Limitations:**

There's no potential negative societal impact of this work in my opinion. The authors discuss the limitations in the checklist.

**Paper Formatting Concerns:**

I haven't noticed any major formatting issues in this paper

**Quality:**

3

**Strengths And Weaknesses:**

### Strengths

1. The paper proved the convergence rate of two-layer ReLU networks on "almost" orthogonal data, with $O(\log(n))$ neurons, I think it's an interesting result.

2. The paper proved the convergence rate by lower-bounding the local-PL curvature, which looks interesting and could technically benefit future works.

3. The paper is in general well-written and is easy to follow the main points. The definition of $r_i(t)$ in Theorem 1 is not clear enough though, I suggest the authors define it properly.


### Weaknesses

1. This paper only discussed the convergence rates of the two-layer ReLU networks, while I believe the results are new and interesting, I'm not sure how significant are the results. I wonder if there are further insights, such as the implicit bias of the  weights, can be induced from the results?

2. The results in Proposition 2 which requires the group initialization are a bit restrictive. In particular, the group initialization requires each neuron to be active with only one different group of data points. This seems to mean that: (1) the weights are already learned some non-trivial information of the data, (2)  since all the data are orthogonal, the weights are moving in different subspaces, and the training of the networks are equivalent to training $p_n$ independent $1$-neuron ReLU network on different datasets.

3. There are also a few other assumptions that limit the scope of the results, such as the requirements of large $d,$  special initialization of $|a_j(0)| \geq \|w_j(0)\|_2,$ and completely orthogonal data for faster rates. However, I believe those assumptions may reflect the broader challenge in developing rigorous theoretical descriptions of neural networks, and I do not consider them as major weaknesses.

---

> ### Author Rebuttal · Authors · 2025-07-31
>
> **Response**: Thank you for the review, we addressed your questions below.
>
> **1. Proposition 2**. Orthogonal data enables us to integrate exactly the system, and have the tight bound in equation (10), of order $\frac{1}{n^{\alpha}}$. We need this tightness to show that all speeds in the interval $[\frac{1}{n}, \frac{1}{\sqrt{n}}]$ can be achieved by appropriate weight initialization.
>
> **2. Group Initialization**. We think that there exists a small enough variance for which the Polyak-Łojasiewicz (PL) curvature still remains of order $\frac{1}{n^{\alpha}}$, but the lack of differentiability of the problem makes this a non-trivial claim which we should further verify. Note that the main point of the group initialization is to show that the speed interval $[\frac{1}{n}, \frac{1}{\sqrt{n}}]$ are achieved by some initialization of the weights.
>
> **3. Implicit Bias**. The problem of the implicit biases are intertwined with our Conjecture: if the weights are approximately equal in magnitude (i.e., implicit bias of an L2-norm), then the speed should be of order $\frac{1}{\sqrt{n}}$; and if the weights are very unequal (i.e., implicit bias of an L1 norm), then the speed will be of order $\frac{1}{n}$. The experiments in section 5.2 suggests a speed of order $\frac{1}{\sqrt{n}}$, and thus an implicit bias minimizing the L2 norm. We did not discuss why convergence speed would be related to implicit bias, and think this would be an interesting direction for future work.
>
> **4. Residual Definition**. Thank you for pointing this out — we will add the definition to the paper. For completeness, we provide it here $r_i(t) = y_i - h_{\theta(t)}(x_i)$.

---

> > ### Comment · Reviewer_5dzT · 2025-08-05
> >
> > Thank you for the reply! I understand the technical necessity of the assumptions, and I also appreciate the theoretical results. Thus I will keep my initial score.

---

### Official Review · Reviewer_6fYV · 2025-07-02

**Clarity:** 3
**Significance:** 3
**Originality:** 4
**Rating:** 5
**Confidence:** 3

**Summary:**

This paper analyzes the convergence properties of one-hidden-layer ReLU networks trained via gradient flow on finite datasets. The main contribution shows that when input data have low correlations (which occurs naturally in high dimensions when d ≳ n²), networks with only O(log n) neurons can achieve global convergence with high probability, regardless of initialization scale. The analysis employs a Polyak-Łojasiewicz (PL) framework to characterize convergence rates, proving exponential convergence at rate 1/n in the general case. For orthogonal data, the authors provide refined analysis showing convergence rates between 1/n and 1/√n, and demonstrate phase transition phenomena in the convergence dynamics. The theoretical results are supported by numerical experiments that validate the predicted scaling behaviors.

**Questions:**

1. **Necessity of assumptions**: How critical is the asymmetric initialization assumption (Assumption 1)? Can you provide more insight into whether this is merely sufficient or actually necessary? Have you explored alternative initialization schemes that might work?

2. **Scaling improvements**: Your experiments suggest the convergence threshold scales as N(d,p) ≃ C(p)d rather than the theoretical √d. Can you comment on the gap between theory and experiments? Is there potential to improve the theoretical bound to linear scaling in d?

3. **Extension to deeper networks**: How might your analysis extend to networks with more than one hidden layer? Are there fundamental obstacles, or is it primarily a technical challenge?

4. **Practical implications**: Given the restrictive dimensionality requirements, can you discuss scenarios where your results might be practically relevant? Are there applications where d >> n² naturally occurs?

5. **Comparison with existing methods**: How does the performance of networks in your regime compare to other approaches (e.g., kernel methods, linear models) that might be more natural in the high-dimensional, low-sample regime?

**Ethical Concerns:**

["NO or VERY MINOR ethics concerns only"]

**Final Justification:**

I would like to keep the same score as the limitations are still prevalent.

**Limitations:**

The authors adequately acknowledge several key limitations:

1. The restriction to shallow networks and square loss
2. The artificial nature of the asymmetric initialization
3. The high-dimensional requirements


However, they could better address:

1. **Computational considerations**: Training in extremely high dimensions (d ≳ n²) has significant computational costs that aren't discussed
2. **Statistical considerations**: In the regime where d >> n, overfitting becomes a major concern that the paper doesn't address
3. **Broader applicability**: More discussion of when the assumptions might be realistic in practice would be valuable

The limitations discussion is reasonable but could be more comprehensive regarding practical deployment considerations.

**Paper Formatting Concerns:**

The paper follows standard formatting guidelines appropriately.

**Quality:**

3

**Strengths And Weaknesses:**

Strengths:
1. **Novel theoretical contribution**: The paper identifies a new regime for neural network convergence that doesn't rely on either the lazy training regime (NTK) or infinite-width assumptions (mean-field). The requirement of only O(log n) neurons is particularly striking and practically relevant.

2. **Rigorous mathematical analysis**: The use of trajectory-wise Polyak-Łojasiewicz analysis is sophisticated and provides sharp convergence rate characterizations. The proofs appear technically sound and the bounds are non-asymptotic.

3. **Natural assumptions**: Unlike previous work requiring carefully tuned initialization scales, the low-correlation assumption (Assumption 2) arises naturally in high dimensions, making the results more broadly applicable.

4. **Comprehensive analysis**: The paper provides both general results and refined analysis for the orthogonal case, including interesting phase transition phenomena that add theoretical depth.

5. **Experimental validation**: The numerical experiments effectively support the theoretical predictions, particularly regarding scaling laws and convergence thresholds.

Weaknesses:
1. **Restrictive dimensionality requirement**: The condition d ≳ n² is quite strong and may limit practical applicability. While the authors show this leads to √d data points on average, this is still restrictive compared to typical machine learning scenarios where n >> d.

2. **Asymmetric initialization assumption**: Assumption 1 requiring |aj(0)| ≥ ||wj(0)|| is somewhat artificial and doesn't align with standard initialization schemes (as the authors acknowledge). While they provide a counterexample when this fails, the necessity of this assumption is questionable.

3. **Limited scope**: The analysis is restricted to one-hidden-layer networks with square loss. Extension to deeper networks or other loss functions is unclear, limiting the broader impact.

4. **Gap between theory and practice**: The orthogonal data assumption, while mathematically convenient, is unrealistic. The general case requires very high dimensions, creating a disconnect from practical scenarios.

5. **Experimental limitations**: The experiments are conducted on relatively small scales (d=100-2000, n≤3500) and don't fully explore the boundary cases where the theory might break down.

---

> ### Author Rebuttal · Authors · 2025-07-31
>
> **Response**: Thank you for your positive review and thoughtful questions. We’re glad you appreciated the paper. Please find our responses to your comments below:
>
> **1. Necessity of assumptions**. The paper shows that the convergence holds with the asymmetric assumption, i.e.for **all** $j, \, |a_j| > ||w_j||$, and that it may not hold **when** $|a_j| < ||w_j||$ for all $j$, as shown in Appendix C.1. This makes this assumption necessary for the result to hold. Now, the assumption-free case makes room for edge cases that look like the proof from Proposition 3. However, these edge cases often require initializations which are unlikely, and thus can be excluded in a reasonning in high probability. While we believe this assumption is unnecessary, we were unable to rigorously prove this within the scope of the present work.
>
> **2. Scaling improvements**. We do think that the gap between Theorem 1 and the experiments can be improved. In the proof of Theorem 1, we need a constraint on the dimension in two critical steps: 1. the lower bound in equation (23) which requires only $d > n$, this is a rank constraint, and cannot be improved with our method, 2. the lower bound between equations (24) and (25) requires the constraint $n \sim \sqrt(d)$, and comes from an upper bound on the norm of the residuals $|r_i(t)|$. The second constraint could be losened up by having a better control on the residuals. In this case, the theoretical scaling would become $d \sim n$.
>
> **3. Extension to deeper networks**. Multi-layer networks are more difficult to control due to the possibility of intermediate layers collapsing before fitting is complete. In our setup, collapse is avoided thanks to Lemma 1 (which prevents $a_j$ from approaching zero) and by controlling the weights $w_j$. However, in deeper architectures or with multi-dimensional outputs, our result no longer holds, as it becomes difficult to simultaneously control multiple sets of weights. Understanding why deep networks avoid collapse in our regime is an interesting and important question, which warrants further investigation and will likely require new techniques for managing groups of neurons.
>
> **4. Practical implications**. Our goal in this paper was to understand convergence of neural networks in the most natural training setup. Unfortunately, this requires unrealistic assumptions on the data, which should be lifted by future work. In practice, since one-layer networks are not really used, it is hard to find examples to compare ourselves with. To illustrate, if we compare our model in terms of parameters-data ratio, and cite Phi-2 with a 519:1 or the ResNet-50 trained on Cifar with a 460:1 ratio. Given the sizes of the models, this is still far from the scaling $d * p \sim n^2\log(n)$. But should Theorem 1 be extended to the empirical scaling $d\sim n$, the ratios would be very close for some architectures, with a $\sim d*\log(d):d$ ratio.
>
> **5. Comparison with existing methods**. While this would certainly be valuable, we did not compare with existing methods in this paper. This is because the theoretical understanding of neural networks remains far from practical setups. We focused our efforts on understanding convergence rather than generalization. Nonetheless, studying test-time behavior and the properties of the minima found is a natural and important direction for our future work.

---

### Official Review · Reviewer_PKJq · 2025-07-03

**Clarity:** 2
**Significance:** 2
**Originality:** 3
**Rating:** 4
**Confidence:** 4

**Summary:**

This paper investigates the training dynamics of a two-layer ReLU neural network in a mean-field setup. When the data matrix is low-correlated (Assumption 2), it is shown that the gradient flow with the regression loss converges to the interpolating solution with order $O(\exp(-t/n))$, where t is the training time and n is the number of samples.  They confirm that Assumption 2 holds when the data dimensionality d satisfies $d\gtrsim n^2$ with high probability. Moreover, for orthogonal data, they provide a more detailed analysis of the convergence speed and conclude that a phase transition exists with respect to the number of neurons, where the convergence speed changes from $1/n$ to $1/\sqrt{n}$.

**Questions:**

Besides what I raised above, I have the following questions:
- Can the framework based on PL curvature be applied to other activation functions? It appears that the lower bound on the curvature (7) is crucial for achieving global convergence, which may be extended by replacing the indicator function with the derivative of other activations.
- What is the primary motivation for employing the mean-field scaling? Since Theorem 1 only requires $p=O(log n)$, the significant difference between the NTK regime and the mean-field scaling may not occur during training.
- Can the framework be extended to the multi-layer networks (for example, by considering the PL curvature with respect to the output layer weights)?

**Ethical Concerns:**

["NO or VERY MINOR ethics concerns only"]

**Final Justification:**

While I have several concerns about the strict assumption and the potential of the PL-curvature framework, the authors' response partially addresses this, especially in dealing with the loss plateau. Unfortunately, the authors have not responded to my last question, but I would like to change my score to accept.

**Limitations:**

The authors properly address the limitations.

**Quality:**

3

**Strengths And Weaknesses:**

**Strength**
- The authors address the challenge of theoretically understanding deep learning training, a significant issue in the field of deep learning. The global convergence analysis using the PL-curvature in the mildly overparameterized mean-field setup is a novel approach, and it achieves a novel global convergence guarantee.
- This paper is well-written. All theoretical results seem solid.

**Weakness**
- One significant limitation of the results is that Corollary 1 requires a huge dimensionality of $d\gtrsim n^2$ (as also mentioned by the authors), which will be violated in practical situations. Can this condition be mitigated by considering a wider network in the same theoretical framework? While Theorem 1 only requires the network width of $O(\log n)$, I wonder if this narrow width restricts the convergence performance of the network.
- The group initialization defined in section 4 seems quite restrictive. Can this setting be applied to the case where we employ the Gaussian initialization?
- The presentation of Theorem 2 seems confusing. For example, the sentence “the transitions happen at each time $t^j=...$”. Moreover, what is the motivation for comparing $t_n^j(\epsilon)$ and $t_n^j(1-\epsilon)$? Perhaps I have any misunderstandings about this part, so it would be helpful if these points could be clarified.
- In Theorem 1, the formal definition of $r_i(t)$ is lacking in the main paper.

---

> ### Author Rebuttal · Authors · 2025-07-28
>
> **Response**: Thank you for your review and interesting questions. We address them below.
>
> **1. Width of the Network**. The large input dimensionality $d$ and the large network width $p$ are two distinct components in our proof. The dimensionality is crucial for ensuring that the weights do not interact too strongly, which allows us to establish the Polyak-Łojasiewicz (PL) lower bounds. The width of the network, on the other hand, is mainly used to guarantee a good initialization—specifically, that for each input $x_i$, there exists some neuron $w_j$ that is positively correlated with it, with high probability. The scaling $p = \log(n)$ primarily arises due to the ReLU non-linearity; different activation functions require different scalings. For instance, non-zero Leaky ReLU requires only a constant number of neurons. That said, increasing the network width could potentially strengthen the results, but doing so would require a different analysis that is compatible with, yet distinct from, the PL framework.
>
> **2. Group Initialization**. The group initialization setup is a toy example intended solely to demonstrate that the lower and upper bounds are tight for orthogonal data. Its purpose is not to model any real-world data distribution. As stated in Claim 5.2 of our experimental section, we expect that Gaussian data should converge at a rate of $\frac{1}{\sqrt{n}}$.
>
> **3. Theorem 2**. We apologize for the lack of clarity in the original statement. Theorem 2 aims to show that under group initialization, the network learns one group at a time in the large-$n$ limit. This is demonstrated by rescaling the loss, which becomes a piecewise constant and decreasing function in the limit. Consequently, within a short interval of order $\log(n)$, the network fully learns a group. We also quantify the width of this transition using the times $t_n^j(\epsilon)$ and $t_n^j(1 - \epsilon)$, following the methodology used in cutoff phenomena for Markov chains. This analysis reveals that *the transition width decreases as $\frac{1}{\log(n)}$*, which is very slow and hence not visible in practice. We will make that discussion clearer in the revised version of the paper.
>
> **4. Residual**. Thank you for pointing this out—we indeed omitted the definition of the residual. We will clarify that $r_i(t) = y_i - h_{\theta(t)}(x_i)$ denotes the $i$-th residual of the loss.
>
> **5. Activation Function**. It is possible to use other activation functions and still obtain convergence results. A simple example is any monomial function of the form $x \mapsto \alpha x^\beta$, with $\beta > 1$ and $\alpha \neq 0$. Such functions satisfy Lemma 1 up to a constant, and the proofs adapt easily. We believe other monotonic activation functions could also work well, though computing their convergence rates might be more challenging. While various activation functions are indeed applicable, we chose not to discuss them in the main text as they do not significantly contribute to the intuition. However, we will include an appendix discussing how different activation functions affect the convergence probability and rate.
>
> **6. Mean-Field Scaling**. We adopted mean-field scaling to normalize the network and to clearly distinguish our regime from the Neural Tangent Kernel (NTK) regime, even as $d$, $n$, or $p$ become large. Using other scalings often hides key constants (e.g., $\frac{1}{p}$) inside initialization terms, as is done in PyTorch, which makes the limit $p \to \infty$ less transparent. We chose this scaling for the sake of clarity and interpretability.
>
> **7. Multi-Layer Networks**. Multi-layer networks are more difficult to control due to the possibility of intermediate layers collapsing before fitting is complete. In our setup, collapse is avoided thanks to Lemma 1 (which prevents $a_j$ from approaching zero) and by controlling the weights $w_j$. However, in deeper architectures or with multi-dimensional outputs, our result no longer holds, as it becomes difficult to simultaneously control multiple sets of weights. Understanding why deep networks avoid collapse in our regime is an interesting and important question, which warrants further investigation and will likely require new techniques for managing groups of neurons.

---

> > ### Comment · Reviewer_PKJq · 2025-08-04
> >
> > I sincerely appreciate the authors' response.
> >
> > The response addresses some of my concerns, but I still have concerns about the high-dimensional assumption, specifically the potential of the PL-curvature-based framework.
> > Particularly, when the learning curve has a plateau, which sometimes happens in the neural network training [1], the lower bound of the PL-curvature will reach zero.
> > In this regard, I wonder if the approach of this paper, establishing a lower bound to the PL-curvature uniformly over the training, could fully explain the training dynamics of neural networks.
> >
> > [1] Ainsworth, Mark, and Yeonjong Shin. "Plateau phenomenon in gradient descent training of RELU networks: Explanation, quantification, and avoidance." SIAM Journal on Scientific Computing 43.5 (2021).

---

> > > ### Author Response · Authors · 2025-08-05
> > >
> > > Thank you for your response. We understand your concern but would like to precise the interaction between the local-PL condition and plateaus.
> > >
> > > The reviewer is right in the fact that a plateau *that does not correspond to a global minimum level set* indicates a failure of the PL condition (or the fact that the PL-constant is super small). Describing these plateaus and what happens during the learning in such phases in an interesting topic **which we do not cover here**.
> > >
> > > We are more interested in convergence to a minimum of the loss and, in this regard, the local-PL condition we develop deals with plateaus by working only trajectory-wise. This means that however close to the local minimum we might get, the local-PL will not reach 0 (contrary to some other notion of PL-condition, [1]), and hence it can be used to prove convergence. In Theorem 1, equation 6, we show that the PL constant is lower bounded by $\frac{C}{n} \min_i |1 - \frac{r_i}{y_i}|$. Now, if we imagine that we initialize the neurons with norm $0< \varepsilon \ll 1$, then  $\min_i |1 - \frac{r_i}{y_i}| \sim \varepsilon$ and you will see plateaus in the training loss of time length $\log(\frac{1}{\varepsilon})$.
> > >
> > > Hence, in **terms of convergence**, such theory can cover plateaus and travel near saddle points.
> > >
> > > [1]: Sourav Chatterjee. Convergence of gradient descent for deep neural networks. arXiv preprint367
> > > arXiv:2203.16462, 2022

---

> ### Comment · Reviewer_PKJq · 2025-08-06
>
> Thank you for your response.
> I now understand that your approach can indeed handle scenarios where plateaus are present.
>
> As an additional question, several works, such as [1] and [2], derive local PL conditions and prove convergence in the teacher-student setting, which you also briefly mention in your conclusion.
> Could you please clarify how your results relate to these works?
>
> [1] Zhou, Mo, Rong Ge, and Chi Jin. "A local convergence theory for mildly over-parameterized two-layer neural network." COLT 2021.
> [2] Chizat, Lenaic. "Sparse optimization on measures with over-parameterized gradient descent." Mathematical Programming 194.1 (2022): 487-532.

---

### Decision · Program_Chairs · 2025-09-17

**Decision:**

Accept (poster)

**Comment:**

This paper identifies a regime of global convergence for gradient flow in one-hidden-layer ReLU networks trained on high-dimensional, finite datasets. The main result establishes an exponential convergence rate of $1/n$ for gradient flows (scaled ODE), under: (Assumption  1) initialization conditions, (Assumption 2) weakly correlated data, and (3) a number of neurons $O(\log n)$, where $n$ is the number of samples. Convergence in this regime follows from the observation that low correlation is typical for sub-Gaussian data in high dimensions $(d = \Omega(n^2))$. The results are further refined to a convergence rate between $1/n$ and $1/\sqrt{n}$ for orthogonal data, with a phase-transition phenomenon in the convergence rate demonstrated along the trajectories. The theoretical predictions are corroborated by experiments that display matching empirical rates.

Reviewers highlighted the paper’s solid analysis within a newly identified regime requiring $O(\log n)$ neurons, along with its technical rigor and clear exposition. Initial concerns regarding the assumptions of the analysis were largely addressed to the reviewers’ satisfaction, with further clarification on the implications, relevance, and applicability of the results. Overall, the paper is recommended for acceptance.